# An instantly fixable and self-adaptive scaffold for skull regeneration by autologous stem cell recruitment and angiogenesis

Gonggong Lu[1,2,3,7], Yang Xu[1,3,7], Quanying Liu[1,3], Manyu Chen[1,3], Huan Sun[1,3], Peilei Wang[1,3], Xing Li[1,3], Yuxiang Wang[1,3], Xiang Li[2], Xuhui Hui[2], En Luo[4], Jun Liu[5], Qing Jiang[1,3], Jie Liang[1,3,6], Yujiang Fan [1,3✉], Yong Sun [1,3✉] & Xingdong Zhang[1,3]

Limited stem cells, poor stretchability and mismatched interface fusion have plagued the reconstruction of cranial defects by cell-free scaffolds. Here, we designed an instantly fixable and self-adaptive scaffold by dopamine-modified hyaluronic acid chelating $Ca^{2+}$ of the microhydroxyapatite surface and bonding type I collagen to highly simulate the natural bony matrix. It presents a good mechanical match and interface integration by appropriate calcium chelation, and responds to external stress by flexible deformation. Meanwhile, the appropriate matrix microenvironment regulates macrophage M2 polarization and recruits endogenous stem cells. This scaffold promotes the proliferation and osteogenic differentiation of BMSCs in vitro, as well as significant ectopic mineralization and angiogenesis. Transcriptome analysis confirmed the upregulation of relevant genes and signalling pathways was associated with M2 macrophage activation, endogenous stem cell recruitment, angiogenesis and osteogenesis. Together, the scaffold realized 97 and 72% bone cover areas after 12 weeks in cranial defect models of rabbit ($\Phi = 9$ mm) and beagle dog ($\Phi = 15$ mm), respectively.

[1] National Engineering Research Center for Biomaterials, Sichuan University, 29# Wangjiang Road, Chengdu, Sichuan 610064, P. R. China. [2] Department of Neurosurgery, West China Hospital, Sichuan University, 37# Guoxue Lane, Chengdu, Sichuan 610041, P. R. China. [3] College of Biomedical Engineering, Sichuan University, 29# Wangjiang Road, Chengdu, Sichuan 610064, P. R. China. [4] State Key Laboratory of Oral Diseases & National Clinical Research Center for Oral Diseases & Department of Oral and Maxillofacial Surgery, West China Hospital of Stomatology, Sichuan University, 14#, 3rd, Section of Renmin South Road, Chengdu, Sichuan 610041, P.R. China. [5] School of Biological Science & Medical Engineering, Southeast University, 2# Sipai Building, Xuanwu District, Nanjing, Jiangsu 210096, P. R. China. [6] Sichuan Testing Center for Biomaterials and Medical Devices, Sichuan University, 29 Wangjiang Road, Chengdu 610064, P. R. China. [7] These authors contributed equally: Gonggong Lu, Yang Xu. ✉email: fan_yujiang@scu.edu.cn; sunyong8702@scu.edu.cn

Reconstruction of large craniofacial bone defects caused by cerebral trauma still remains highly challenging[1,2]. Currently, reconstructive operations using autologous/allogeneic grafts suffer from the high cost of bone harvest, limited bone sources, and potential donor site complications[3]. Abundantly available substitutes, including titanium mesh and polyether ether ketone, face drawbacks such as poor stretchability, weak osteointegration, obvious tissue friction, and high Young's modulus, which tend to constrain intracranial tissue and cannot be implanted immediately after craniectomy[4]. Moreover, accessory devices in traditional cranioplasty, e.g., skull locks or bone nails, increase the complexity and cost of surgery. Therefore, a functional implant that could be implanted and fixed in defect sites immediately after craniectomy might bring new prospects to adapt to the intracranial microenvironment and mobilize endogenous stem cells (ESCs) for regeneration of cranial tissue.

Craniofacial bones are formed mainly through intramembranous ossification, in which the mesenchymal cells differentiate directly into osteoblasts, and subsequently develop into an organic/inorganic hybrid cross-linked cancellous bone structure[5]. The reemerging structure and composition of craniofacial bones might provide necessary environmental cues to create an appropriate niche for the migration, infiltration, proliferation, and differentiation of stem cells[6–8]. The formation of cranial bone coincides with capillary evolvement, which generates highly vascularized tissue for the timely supply of oxygen and nutrients to maintain skeletal development, integration, and homeostasis[9,10]. Namely, a close connection between osteogenesis and angiogenesis is crucial in cranial bone formation[11,12]. However, current cranial bone regeneration strategies often fail to produce highly vascularized new bones[13]. The major drawbacks of these strategies might be structural instability, low adhesion with surrounding host tissues in a wet environment, and limited functional cell settlement[14–16].

Hydrogel-based adhesives have been used for sealing tissues or coating implants to improve their adhesion with surrounding tissues and increase the invasion and retention of functional cells[17–20]. Nevertheless, the lack of adhesion in wet environments and inefficient cell recruitment hindered the successful implementation of hydrogel-based adhesives. Meanwhile, several strategies have been proposed to promote angiogenesis, including the structural optimization of biomimetic scaffolds[21], delivery of angiogenic growth factors such as vascular endothelial growth factor (VEGF)[22], or the use of highly potent cell sources such as stem cells or mature vascular cells[23]. However, the ex vivo expansion of exogenous stem cells and in vivo delivery of growth factors are restricted by the limited availability of stem cell sources, excessive cost of commercialization, anticipated difficulties of clinical translation, and regulatory approval. The most effective and feasible strategy for overcoming these problems is to exploit instantly fixable scaffolds that reflected the niche of osteogenic organisms and interacted with the surrounding host tissue to achieve rapid homeostasis by recruiting ESCs, and promoting angiogenesis and osteogenesis.

Inspired by the excellent adhesion ability of mussels in moist or liquid environments, dopamine has been widely considered an important adhesion functional group[24,25]. In this work, dopamine-modified hyaluronic acid (HAD) is employed as a "bridge" to chelate $Ca^{2+}$ of micron hydroxyapatite (µHAp) and bind type I collagen (Col I) through a Michael addition reaction. This strategy effectively integrates organic and inorganic phases with strong chemical connections at the molecular level to form a hybrid cross-linked scaffold (HCLS), which mimics the construction of a natural bony extracellular matrix (Fig. 1). Importantly, it can flexibly deform to respond to the stress from the defect, maintain structural integrity, and present good mechanical match and interface integration with the surrounding bone by

tissue adhesion. Furthermore, the proper immune microenvironment provided by HCLS regulates M2 phenotype polarization of macrophages. After implantation in cranial defect models of both rabbit and beagle dog, HCLS can rapidly initiate angiogenesis and osteogenesis by in-situ recruitment of ESCs, and subsequently, accelerate osteodifferentiation to regenerate the skull.

## Results

**Engineered self-adhesive and flexible porous HCLS.** An HCLS (Fig. 2a) that mimics the composition and structure of the natural skull was developed from dopamine-modified HAD, Col I, and µHAp slurry. The HAD, Col I, and their dual cross-linked scaffold (DCLS) were prepared for comparison. µHAp was microhydroxyapatite particles with an average size of approximately 14.45 µm and a Ca/P ratio of 1.67, as confirmed by X-ray diffraction (XRD), dynamic light scattering, scanning electron microscopy (SEM), and energy dispersive spectroscopy (EDS) analysis (Supplementary Fig. 1a–d). The substitution degree of the catechol group in HAD was 7.8% based on $^1H$ NMR (Supplementary Fig. 2a, b). The strong absorption peak at 277 nm in the UV spectrum indicated the catechol structure (Supplementary Fig. 2c)[26]. In the FTIR spectrum of HAD (Supplementary Fig. 2d), the peaks at 1553 cm$^{-1}$, and 1151 cm$^{-1}$ demonstrated the successful amidation[27], and the peak at 1731 cm$^{-1}$ represented phenolic hydroxyl groups. In the HCLS spectrum, the 1731 cm$^{-1}$ peak disappeared, indicating the reaction of DOPA with inorganic µHAp through phenol-quinone transition and Michael addition between quinone and amino groups of Col I, which proved that HAD chemically cross-linked with µHAp and Col I molecules rather than being physically mixed[28]. The DSC results (Fig. 2b) revealed that HCLS achieved the highest denaturation temperature of 113.5 °C relative to 84.6 °C of Col I, 88.2 °C for HAD, 111.7 °C for DCLS, and 81.6 °C for HA. The denaturation temperatures of HA-Col I-µHAp and HA-Col I without the introduction of dopamine were 100.6 °C and 94.7 °C, respectively, suggesting that the hybrid cross-linking strategy enhanced the heat stability of HCLS. There was no additional weight loss phase in HCLS, DCLS, and Col I in comparison with HA-Col I and HA in the thermogravimetric analysis curves, and the content of µHAp in the HCLS group was 19% w/w (Fig. 2c).

To further demonstrate the structural superiority of HCLS at the molecular level, X-ray photoelectron spectroscopy (XPS) and SEM analysis were performed to reveal chemical bonding and calcium chelation. The chemical bonding between HAD and Col I was investigated by XPS (Fig. 2d1 and Supplementary Fig. 3a). Higher -C=O (284.3 eV, 62.2% and 530.8 eV, 45.7%) and lower -C-OH peak areas (286.2 eV, 30.4% and 532.6 eV, 19.2%) were observed in DCLS than in HAD (-C=O, 284.3 eV, 38.5% and 530.8 eV, 23.1%; and -C-OH, 286.2 eV, 60.8% and 532.6 eV, 72.3%), indicating that HAD was oxidized and bound with amino groups in Col I. Furthermore, the relatively more -C=O groups in DCLS (284.3 eV, 62.2%) suggested that the phenolic hydroxyl groups in HAD were converted to quinone groups during oxidation, which might provide more nucleation sites and lead to the nucleation and growth of hydroxyapatite crystals. Calcium chelation analysis in HCLS demonstrated an obvious Ca2p peak (Ca2p3/2, 346.5 eV), which was similar to that in HAp (Ca2p3/2, 346.8 eV). However, the P2p peak in HAp (P2p, 132.9 eV) disappeared in HCLS, which suggested that $Ca^{2+}$ originating from HAp was ionized to form chelated calcium (coordinated calcium) with the cross-linked DCLS polymer (Fig. 2d2 and Supplementary Fig. 3b).

To confirm the chelation capacity of calcium ions, DCLS and HCLS were treated with 0.3 M $CaCl_2$ and characterized by XPS

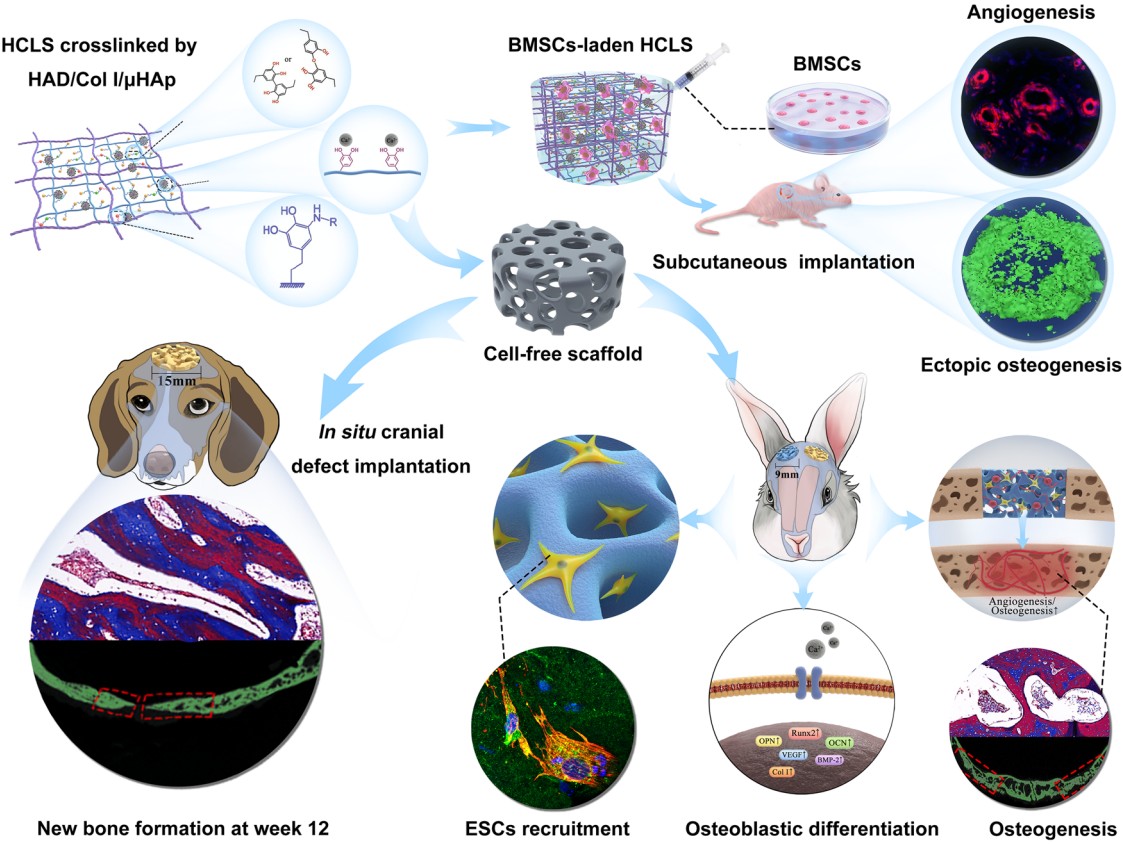

**Fig. 1 Clinical treatment of cranial defects by cell-free scaffolds faced significant challenges and imperious demands.** Biomimetic dopamine-mediated hybrid cross-linked realized skull reconstruction through instantly seamless adhesion at defect area, and recruitment and retention of endogenous stem cells to rapid initiation of osteogenesis and angiogenesis. After 12 weeks implantation, the HCLS could realize extensive bone regeneration with bone cover area at 97% for rabbit model ($\Phi = 9$ mm) and 72% for beagle dog model ($\Phi = 15$ mm) at cranial defects sites.

(Fig. 2d3 and Supplementary Fig. 4a). The obvious increase in the Ca2p peak (Ca2p1/2, 351.0 eV and Ca2p3/2, 347.5 eV) in HCLS-0.3 M $Ca^{2+}$ indicated the sustained calcium ion chelation ability of HCLS, even after chelating ionized calcium derived from HAp (Fig. 2d3). In addition, more $-C = O$ (284.3 eV and 531.3 eV) was found in comparison with $-C-OH$ (286.2 eV and 532.9 eV) in both DCLS and HCLS, implying that catechol groups were oxidized to quinones which could further chelate with $Ca^{2+}$ (Supplementary Fig. 4b, c). In vitro mineralization experiments were carried out to further verify the chelation capacity of calcium ions in HCLS (Supplementary Fig. 5). SEM images revealed that the microspherical osteoid apatite agglomerated on the surface of HCLS; high-magnification SEM images revealed that all the osteoid apatite particles had a nanolath-like structure, a typical hydroxyapatite crystal (Supplementary Fig. 5a)[29]. EDS analysis demonstrated that the crystals were mainly composed of Ca and P with a Ca/P ratio of 1.74, which was close to mineralized hydroxyapatite in natural bone (1.67) (Supplementary Fig. 5b)[30].

Col I was distributed evenly in HCLS (Fig. 2e1) as verified by Masson's trichrome staining, and µHAp was distributed uniformly as confirmed by calcein staining using confocal laser scanning microscopy (CLSM) (Fig. 2e2)[6,31,32]. The high porosity ($87 \pm 9\%$) and pore size ($359 \pm 49$ µm) similar to those for natural cancellous bone were observed by SEM (Fig. 2f1–3 and Supplementary Fig. 6a). Compared with obvious swelling of HAD ($175 \pm 10\%$) and contraction of Col I ($26 \pm 5\%$), there was no significant size change in the HCLS ($95 \pm 8\%$) and DCLS groups ($86 \pm 9\%$), even after 14 days of incubation in Dulbecco's phosphate-buffered saline (DPBS) in vitro (Fig. 2g and Supplementary Fig. 7). Accordingly, the mechanical strength of HAD

and Col I decayed rapidly, but the cross-linked gels, especially the HCLS hydrogel, maintained an almost constant storage modulus (G′, Fig. 2h), suggesting that the incorporation of µHAp increased the cross-linking degree by the newly formed chelation of $Ca^{2+}$ derived from µHAp to strengthen the internal network structure. This conjecture was consistent with the visual SEM results. The enzymatic degradation test further verified the relatively good structural stability of HCLS (Supplementary Fig. 6b).

HCLS achieved the highest tensile strength of 34.8 KPa at the breakdown point, and it could be folded like origami and unfolded into a flat sheet (Figs. 2i1, 3 and Supplementary Video 1). The mechanical properties of the scaffolds with blood infiltration (simulating the in vivo application environment) were further evaluated. Although the tensile forces of the scaffolds decreased dramatically from 4.2 to 0.3 N for DCLS and 5.6–0.7 N for HCLS, respectively, with blood infiltration, the deformation range was obviously elongated (Fig. 2j, Supplementary Fig. 8a). Dynamic Mechanical Analyzer (DMA) test results indicated that the storage modulus (G′) of DCLS declined markedly after blood infiltration (55–35 KPa), but no significant differences in G′ were observed in HCLS (90–85 KPa) (Fig. 2k1–3, Supplementary Fig. 8b). The HCLS could be instantly fixed to the cranial defect site in a wet environment, probably owing to the high affinity of catechol groups for diverse nucleophiles (e.g., amines, thiols, imidazoles, and chelation of metal particles) (Fig. 2i2 and Supplementary Video 2), which provided a potential adhesion mechanism by anchoring to proteins on the skull surface[33]. As demonstrated in Fig. 2l and Supplementary Fig. 9, the adhesion strength of HCLS was the highest against the glass, titanium (Ti), skull, pigskin, and polyethylene.

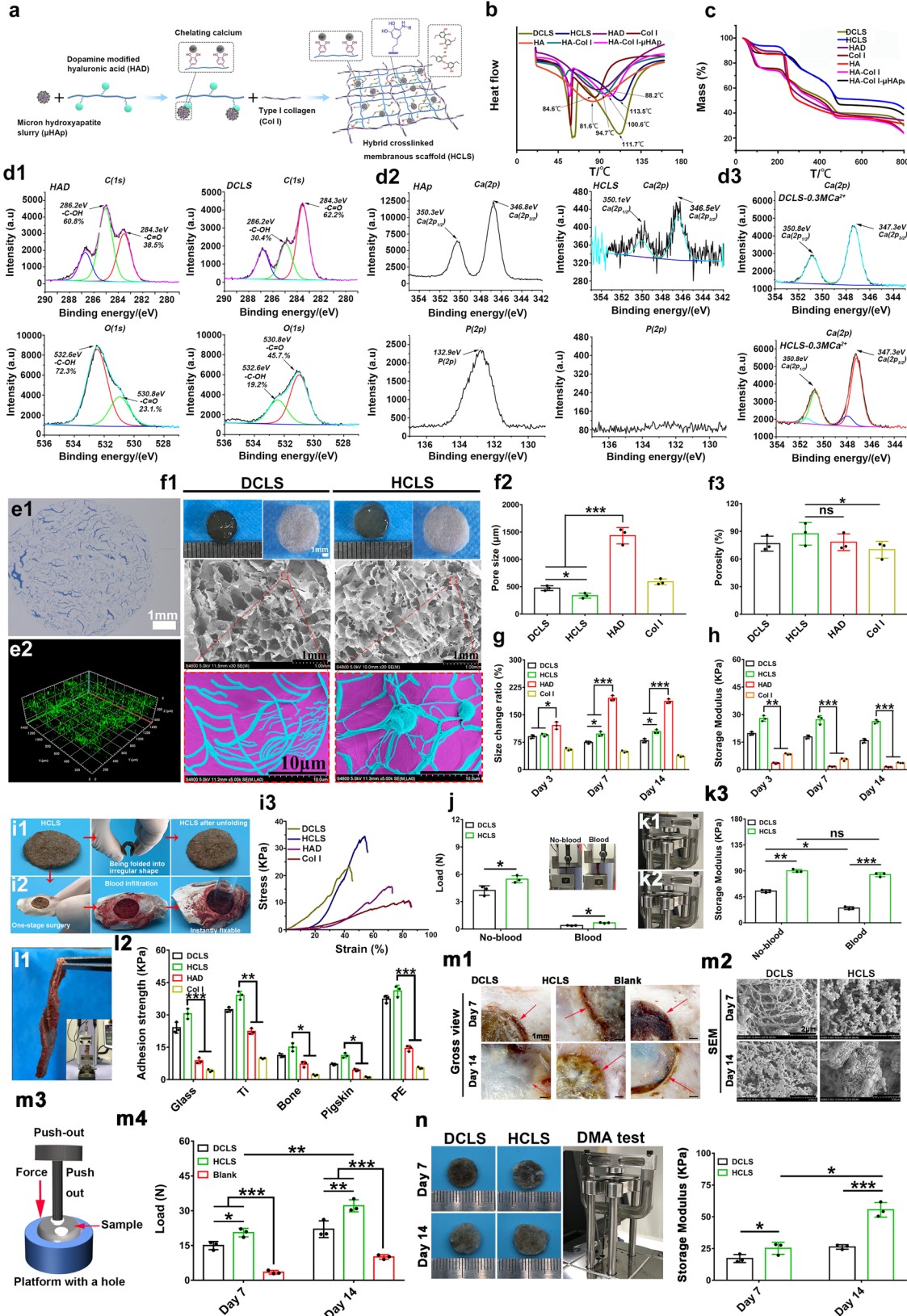

The interfacial bonding between scaffolds and host bone was also evaluated in rabbit skull defect model ($\Phi = 9$ mm) after 7 and 14 days of implantation. No obvious size change occurred after 14 days' implantation. The interface was well-integrated on day 7, and became more tightly connected on day 14 by gross observation and SEM (Fig. 2m1 and Supplementary Fig. 10).

A more obviously mineralized fiber structure in HCLS was observed on day 7; on day 14, the mineralized matrix became more intense and nanolath-like osteoid apatite particles appeared, which was consistent with the results of in vitro mineralization (Fig. 2m2). The Ca/P ratios and element analysis results further confirmed that the minerals in HCLS were closer to natural HAp

**Fig. 2 Engineered self-adhesive and flexible porous hybrid cross-linked scaffold. a** Schematic of the synthesis and structure of HCLS. **b** DSC test of various specimens. **c** TG analysis of various specimens. **d1** High-resolution X-ray photoelectron spectroscopy spectra of C1s and O1s in HAD and DCLS. **d2** High-resolution x-ray photoelectron spectroscopy spectra of Ca2p and P2p in HAp and HCLS. **d3** High-resolution X-ray photoelectron spectroscopy spectra of Ca2p in DCLS-0.3MCa$^{2+}$ and HCLS-0.3MCa$^{2+}$. **e1** Representative Masson's trichrome staining in HCLS group. **e2** CLSM image showing the distribution of μHAp inside the HCLS by calcein staining. **f1–3** SEM images of various scaffolds and corresponding pore size and porosity (*$p = 0.0202$, ***$p = 0.0004$, ns $= 0.1384$, *$p = 0.0496$). **g** Quantitative analysis of swelling test (*$p = 0.0102$, *$p = 0.0232$, *$p = 0.0335$, ***$p = 6.9215 \times 10^{-5}$, ***$p = 1.2657 \times 10^{-6}$). **h** DMA test at 5 Hz after 3, 7, and 14 days swelling (***$p = 3.0101 \times 10^{-4}$, ***$p = 8.6268 \times 10^{-7}$, ***$p = 5.428 \times 10^{-6}$). **i1** Flexible HCLS with foldability. **i2** Instant fixability of HCLS. **i3** Tensile test of various scaffolds. **j** Tensile test of various scaffolds with or without invasive blood (*$p = 0.0116$, *$p = 0.0314$). **k1–3** Characterization of compressive storage modulus with or without invasive blood (**$p = 0.0027$, ***$p = 0.0005$, *$p = 0.0202$, ns $= 0.0552$). **l1, l2** Adhesion test of various hydrogels on different substrate (***$p = 2.4921 \times 10^{-6}$, **$p = 0.0023$, *$p = 0.0410$, *$p = 0.0262$, ***$p = 7.8344 \times 10^{-7}$). **m1, m2** Gross view and SEM images of mineralization at the interface between implant and host bone on day 7 and 14. **m3, m4** Push-out test of various samples after 7 and 14 days's implantation (*$p = 0.0190$, *$p = 0.0171$, **$p = 0.0034$, ***$p = 1.8517 \times 10^{-5}$, ***$p = 0.0002$). **n** DMA test of various implants at day 7 and 14 (*$p = 0.0445$, ***$p = 2.6672 \times 10^{-5}$, *$p = 0.0103$). $n = 3$ independent replicates from three samples. (Two-sided comparison, error bars represent standard deviation, *$p < 0.05$, **$p < 0.01$, and ***$p < 0.001$).

(Supplementary Fig. 12). These results suggested that HCLS had a strong calcium-binding ability, which could induce HAp formation to facilitate good osseointegration. The interfacial binding force increased over time, and the highest push-out force was achieved in HCLS on Day 14 ($32 \pm 3$ N) (Fig. 2m3, m4 and Supplementary Fig. 11a). The G′ of implants increased to $60 \pm 3$ KPa in HCLS on Day 14 (Fig. 2n and Supplementary Fig. 11b), demonstrating satisfactory structural stability and interface integration.

**The immune response of HCLS in vitro and in vivo.** First, the immune responses of various scaffolds were evaluated in vitro. Compared with the smooth boundaries of RAW 264.7 macrophages on DCLS, abundant filopodia with long stretching distances were found on HCLS by SEM. The quantitative results showed a higher spreading area and cell aspect ratio on HCLS (Supplementary Fig. 13a, c1, c2), implying the promotion of cell spreading and distinctive cell morphologies by μHAp. Immunofluorescent staining was applied to visually analyze the immunological polarity induced by HCLS and DCLS (Supplementary Fig. 13b). The semiquantitative immunofluorescence intensity by Image J showed that HCLS exhibited higher CD206$^+$ (highly specific M2 type marker) and lower CD197$^+$ expression (highly specific M1 type marker) than DCLS (Supplementary Fig. 13c3, c4).

H&E analysis showed that infiltration of inflammatory cells was mainly concentrated on the periphery of both DCLS and HCLS on Day 7 after intramuscular implantation in mice. Fibrous tissue was found penetrating into the interior zone of the implants on Day 14 (Fig. 3a and Supplementary Fig. 14a). A large number of cells were observed both in HCLS and DCLS, but the cells in HCLS presented a higher spreading area and cell aspect ratio than those in DCLS (Fig. 3b and Supplementary Fig. 14b1 and b2). Flow cytometry data indicated that the macrophage levels in HCLS were lower than those in DCLS, and decreased over time (Fig. 3c1, c2). HCLS recruited higher numbers of M2 macrophages (CD197$^-$CD206$^+$ cells) and fewer M1 macrophages (CD197$^+$CD206$^-$ cells) (Fig. 3c3, c4). The immune-related cytokines in vivo were examined using ELISA to further investigate the inflammatory responses (Fig. 3d1–5). Compared with DCLS, HCLS downregulated proinflammatory cytokines (TNF-α and IL-1β), and upregulated anti-inflammatory cytokines (IL-4, IL-10, and IL-1rα), and this effect became more obvious over time. Immunofluorescence staining further confirmed that HCLS was more conducive to the M1-to-M2 shift of macrophages than DCLS (Fig. 3e1–3 and Supplementary Fig. 17). Macrophages were spatially distributed around the implants on day 7, whereas more macrophages penetrated into the implants on day 14. (Supplementary Fig. 16).

Macrophages can mediate bone and vascular regeneration by secreting osteoinductive and angioinductive factors[34,35], such as BMP-2 and VEGF as potent inducers of osteogenesis and angiogenesis[36,37]. Immunofluorescent staining was applied to explore whether HCLS contributed to osteogenesis and angiogenesis (Fig. 3e1, e4, e5 and Supplementary Fig. 17). It was found that most endogenous VEGF and BMP-2 colabelled with F4/80$^+$ macrophages appeared in the implants. The expression level increased over time both in DCLS and HCLS, but the fluorescence intensity in HCLS was obviously higher than that in DCLS. A similar phenomenon occurred in the rabbit skull defect model (Fig. 3f1–5 and Supplementary Figs. 18 and 19). Notably, HCLS presented stronger proinflammatory properties (CD197$^+$CD206$^-$) 7 days after implantation, but inflammation was significantly reduced on day 14. It was reported that the inflammation at the early stage of wound repair benefited the recruitment of inflammatory cells, biochemical factors, and bone progenitor cells[38]. Hence, higher BMP-2 and VEGF secretion induced by HCLS could be attributed to polarized M2 phenotype macrophages[39,40].

**Osteogenic differentiation of BMSC-laden HCLS in vitro and in nude mouse subcutaneous implantation.** The bone marrow stromal cells (BMSCs) were encapsulated into hydrogels and cultured for 14 days to evaluate their proliferation and differentiation. Live/dead staining (Fig. 4a1 and Supplementary Fig. 20a) indicated that HAD promoted the uniform spread of cells, the introduction of collagen increased cell adhesion, and the cross-linked μHAp in HCLS improved visual cell density. The CCK-8 results confirmed that all the hydrogels could promote BMSC proliferation to varying degrees, while the HCLS group performed the best (Fig. 4a2). CLSM images of cytoskeletal staining and SEM images (Fig. 4b1, b2, bc and Supplementary Fig. 20b, c) of BMSCs showed that the introduction of dopamine increased cell adhesion and spread, with highly elongated actin filaments (red) surrounding the cell nuclei (blue), especially for HCLS. Interestingly, the immunohistochemical staining of VEGF and BMP-2 demonstrated that a larger area of positive staining was evenly distributed in the HCLS group (Fig. 4d1, e and Supplementary Fig. 20d); the semiquantitative results verified the optimal BMP-2 secretion (Fig. 4d2). Furthermore, the specific osteogenic gene expression levels of Runx2, osteocalcin (OCN), and osteopontin (OPN) were gradually upregulated over time, and the related expression level in the HCLS was significantly higher than those in the other groups (Fig. 4f1–3).

To simulate bone matrix deposition and mineralization, BMSC-laden hydrogels (including DCLS, HCLS, HAD, and Col I) were subcutaneously implanted into nude mice for 30 days (Fig. 4g). The size change ratio of 0.95 demonstrated the structural stability

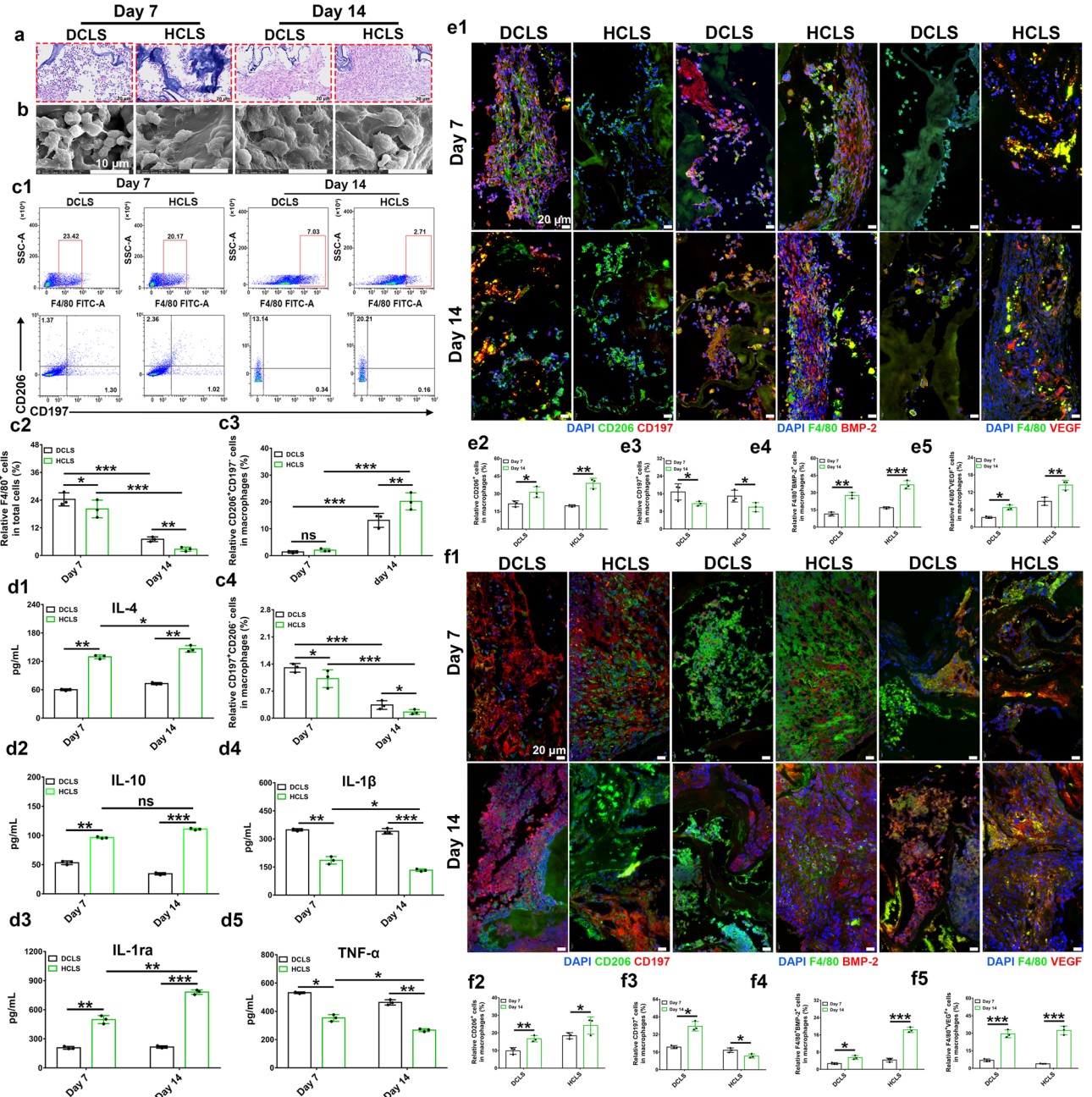

**Fig. 3 In vivo immune response of scaffold in mouse intramuscular and rabbit skull defect model. a** H&E staining of implants on days 7 and 14 after implantation in mouse intramuscular. **b** Cell morphologies of macrophages on different implants. **c1**) Representative flow cytometry plots of F4/80+, CD197+CD206− macrophages (M1) and CD197−CD206+ macrophages (M2) polarization in DCLS and HCLS on days 7 and 14 after implantation in mouse intramuscular. **c2–4**) Relevant quantification of F4/80+, CD197+CD206− macrophages (M1) and CD197−CD206+ macrophages (M2) polarization in DCLS and HCLS (**c2**: **$p = 0.0054$, ***$p = 0.0004$, ***$p = 0.0009$, *$p = 0.0431$; **c3**: **$p = 0.0078$, ***$p = 2.1538 \times 10^{-5}$, ***$p = 0.0006$, ns = 0.1301; **c4**: **$p = 3.7781 \times 10^{-5}$, ***$p = 6.6342 \times 10^{-5}$, *$p = 0.0482$, *$p = 0.0365$). **d1–5** Expression of inflammatory factors induced by implants after implantation in the mouse intramuscular model for 7 and 14 days. $n = 3$ biologically independent replicates (**d1**: *$p = 0.0378$, **$p = 0.0018$, **$p = 0.0022$; **d2**: **$p = 0.0065$, ***$p = 7.8921 \times 10^{-5}$, ns = 0.0843; **d3**: **$p = 0.0024$, **$p = 0.0019$, ***$p = 2.4562 \times 10^{-5}$; **d4**: *$p = 0.0342$, **$p = 0.0055$, ***$p = 4.1243 \times 10^{-7}$; **d5**: *$p = 0.0158$, *$p = 0.0301$, **$p = 0.0087$). **e1** Representative immunostainings of CD197 and CD206, BMP-2 secretion and F4/80+, and VEGF secretion and F4/80+ macrophages on days 7 and 14 after implantation in mouse intramuscular. **e2–5** Relevant semiquantitative analysis of immunofluorescence staining by Image J software **e2** *$p = 0.0180$, **$p = 0.0091$; **e3**: *$p = 0.0342$, *$p = 0.0435$; **e4**: **$p = 0.0033$, ***$p = 8.0122 \times 10^{-6}$; **e5**: *$p = 0.0185$, **$p = 0.0044$). **f1** Representative immunostainings of CD197 and CD206, BMP-2 secretion and F4/80+ and VEGF secretion and F4/80+ macrophages on days 7 and 14 after implantation in rabbit skull defect. **f2–5** Relevant semiquantitative analysis of immunofluorescence staining by Image J software (**f2**: **$p = 0.0020$, *$p = 0.0416$; **f3**: *$p = 0.0196$, *$p = 0.0394$; **f4**: *$p = 0.0149$, ***$p = 5.5420 \times 10^{-8}$; **f5**: ***$p = 4.6347 \times 10^{-6}$, ***$p = 8.1728 \times 10^{-7}$). $n = 3$ cells examined three independent experiments. (Two-sided comparison, error bars represent standard deviation, *$p < 0.05$, **$p < 0.01$, and ***$p < 0.001$).

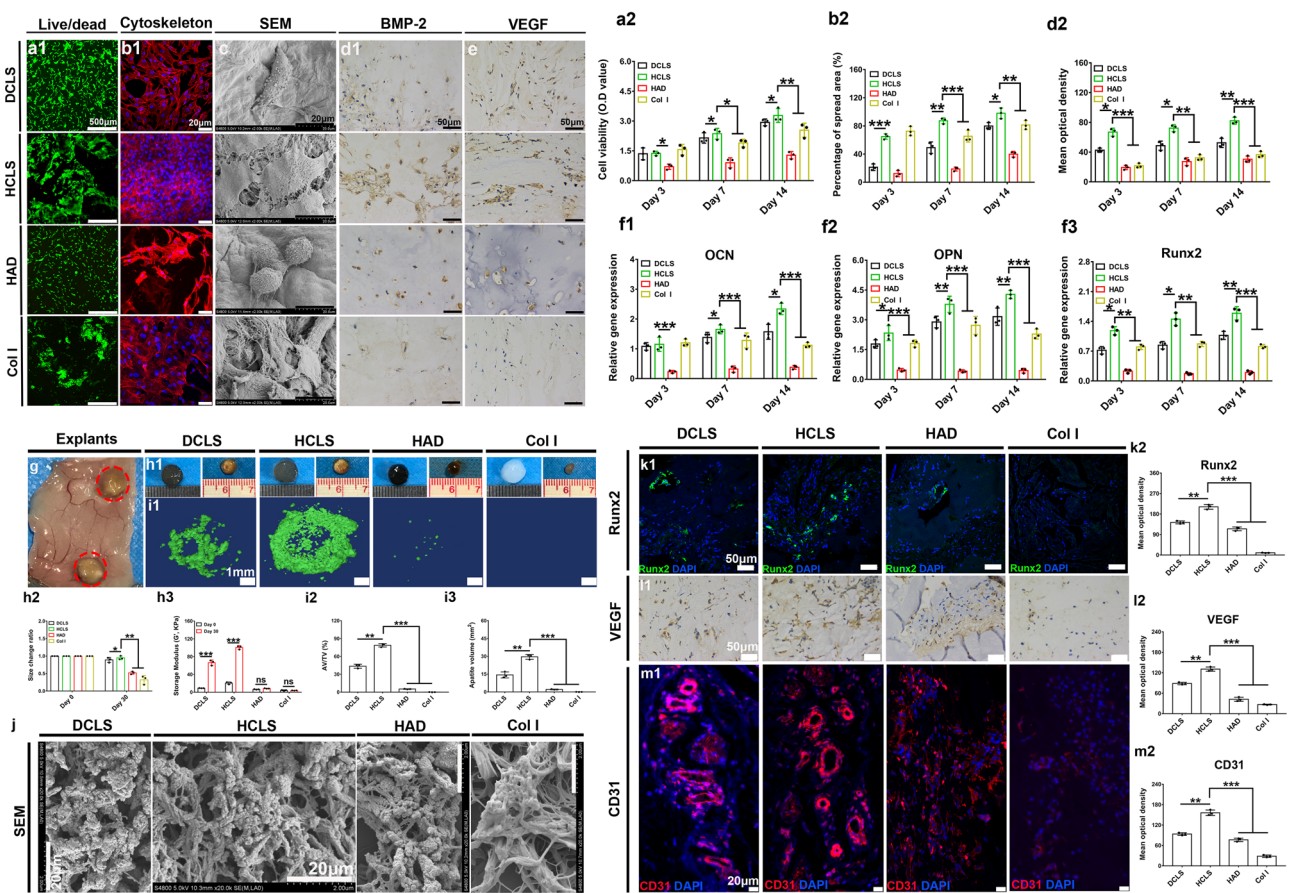

**Fig. 4 In vivo osteodifferentiation and ectopic mineralization of BMSCs-laden HCLS. a1** CLSM images (Live/dead staining) of BMSCs-laden various hydrogels for 14 days. **a2** Cell viability of BMSCs-laden various hydrogels at day 3, 7 and 14 (*$p = 0.0140$, *$p = 0.0335$, *$p = 0.0123$, *$p = 0.0453$, ***$p = 1.9112 \times 10^{-7}$). $n = 3$ biologically independent replicates. **b1** CLSM images (rhodamine-phalloidin/DAPI staining) of BMSCs on day 14. **b2** Percentage of cell spreading area after 3, 7, and 14 days (**$p = 0.0022$, ***$p = 8.0407 \times 10^{-7}$, ***$p = 3.1642 \times 10^{-6}$, **$p = 0.0012$, *$p = 0.0206$). **c** SEM images of BMSCs distributed on gel scaffolds after 14 days incubation. **d1, d2**) Immunohistochemistry staining of BMP-2 at day 14 and semiquantitative analysis of BMP-2 staining by Image J software (***$p = 8.0447 \times 10^{-5}$, ***$p = 1.0591 \times 10^{-4}$, **$p = 0.0032$, **$p = 0.0026$, *$p = 0.0124$, *$p = 0.0221$). $n = 3$ cells examined three independent experiments. **e** Immunohistochemistry staining of VEGF on day 14. **f1**–3 Gene expression of OCN, OPN and Runx2 for the BMSCs encapsulated in various hydrogels at day 3, 7 and 14. All gene expressions were normalized to housekeeping gene β-actin (**f1**: ***$p = 0.0004$, ***$p = 6.0578 \times 10^{-6}$, ***$p = 1.2233 \times 10^{-7}$, *$p = 0.0156$, *$p = 0.0147$; **f2**: *$p = 0.0464$, **$p = 0.0067$, **$p = 0.0014$, ***$p = 9.8374 \times 10^{-7}$, ***$p = 1.3141 \times 10^{-9}$, ***$p = 2.8964 \times 10^{-10}$; **f3**: *$p = 0.0461$, **$p = 0.0031$, *$p = 0.0139$, **$p = 0.0069$, **$p = 0.0023$, ***$p = 9.2436 \times 10^{-6}$). $n = 3$ biologically independent replicates. **g** Visualization subcutaneous state after 30 days implantation. **h1** Gross appearance of the samples before and after implantation. **h2**) Size change ratio of samples before and after implantation (*$p = 0.0448$, **$p = 0.0056$). **h3** Dynamic mechanical test after 30 days implantation (***$p = 3 \times 10^{-15}$, ***$p = 1 \times 10^{-15}$, ns = 0.3906, ns = 0.9007). **i1** 3D reconstruction images of various explants by Micro-CT. **i2, i3** Quantitative bone volume analyses (AV: apatite volume. TV: total volume.) **i2**: **$p = 0.0018$, ***$p = 6.7311 \times 10^{-9}$; **i3**: **$p = 0.0023$, ***$p = 5.8921 \times 10^{-9}$). **j** SEM images of the inner section of various samples. $n = 3$ biologically independent replicates. **k1** Immunofluorescence staining to detect Runx2 in various BMSCs-loaded hydrogels. **k2** Semiquantitative analysis of Runx2 staining by Image J software (**$p = 0.0066$, ***$p = 0.0002$). **l1** Immunohistochemistry staining to detect VEGF in BMSCs-loaded hydrogels. **l2** Semiquantitative analysis of VEGF staining by Image J software (**$p = 0.0004$, ***$p = 5.7513 \times 10^{-5}$). **m1** CD31 staining of various explants. **m2** Semiquantitative analysis of CD31 staining by Image J software (**$p = 0.0073$, ***$p = 0.0001$). $n = 3$ cells examined three independent experiments. (Two-sided comparison, error bars represent standard deviation, *$p < 0.05$, **$p < 0.01$, and ***$p < 0.001$).

of BMSC-laden HCLS in vivo (Fig. 4h1, h2). Interestingly, the storage modulus of HCLS ($103 \pm 8$ kPa) was nearly five times its initial value (Fig. 4h3, also relative to $65 \pm 4$ kPa, $7 \pm 2$ kPa, and $4 \pm 0.7$ kPa for DCLS, HAD, and Col I, respectively), implying abundant bone matrix deposition and mineralization inside the HCLS. The 3D reconstruction of micro-CT images further demonstrated larger amounts of mineralized bone-like deposition in HCLS (Fig. 4i1, Supplementary Fig. 21a). Quantitative analysis of the apatite volume (AV) and apatite volume relative to the total volume (AV/TV) confirmed that more apatite was deposited in HCLS ($30 \pm 2.3$ mm$^3$ and $78 \pm 4.1\%$) (Fig. 4i2, i3). The SEM images of explants showed porous structures and entangled

fibrous networks (Fig. 4j and Supplementary Fig. 21b). In particular, some mineral nanoparticles with an average size of approximately 200 nm were evenly embedded into the entangled fibrous bundle in HCLS, which was highly similar to the natural bone matrix structure[41]. The EDS result of HCLS (Supplementary Fig. 21c) revealed a uniform and dense distribution of Ca and P elements with a Ca/P ratio of $1.91 \pm 0.34$, which was closest to that of natural bone tissue (1.67). The XRD results showed that the peaks at 26° and 32° corresponded to the (002) and (211) diffraction peaks of μHAp, respectively (Supplementary Fig. 21d)[42]. Runt-related transcription factor 2 (Runx2) is a specific transcription factor essential for bone formation that

regulates BMSCs to differentiate into osteoblasts, inhibits their differentiation into chondrocytes and adipocytes, and upregulates the expression of matrix genes (OCN, Col I, etc.)[43]. Positive Runx2 was observed in the DCLS, HCLS, and HAD groups by immunofluorescence staining (IF) (Fig. 4k1, Supplementary Fig. 22a). The semiquantitative analysis further demonstrated the highest expression of Runx2 in the HCLS group (Fig. 4k2), reflecting its osteoinductive potential.

Bone defect healing is a dynamic progenitor cell-driven tissue morphogenetic process that requires coordinated osteogenesis and angiogenesis at the repair site[44]. Hence, the success of bone regeneration also depends on vascularization. Interestingly, visible blood vessels surrounded the explant in Fig. 4g, and obvious positive staining of VEGF and endothelial cell marker (CD31) was observed (Fig. 4l1, l2; Supplementary Fig. 22b and Fig. 4m1, m2), suggesting that new blood vessel formation might be closely related to HAD[45]. Moreover, the significantly enhanced positive expression of VEGF and CD31 in HCLS implied that the introduction of μHAp accelerated angiogenesis.

**Endogenous stem cell recruitment by HCLS.** It would be significantly valuable if the HCLS could actively recruit endogenous stem cells (ESCs). Here, first, a BMSC suspension was seeded onto the surface of lyophilized scaffolds and soaked in a culture medium for 3, 7, and 14 days. Subsequently, the scaffolds were rinsed thoroughly to remove nonadherent cells. Next, the live/dead and cytoskeleton staining were performed. Interestingly, the number and spreading area of adhered cells in HCLS increased significantly in comparison to other groups (Supplementary Fig. 23a, b); the typical morphology of BMSCs with fusiform shape and intensive adhesive contraction illustrated the effective adhesion behavior of HCLS (Supplementary Fig. 23c). These phenomena indicated that HCLS could better anchor BMSCs and promote proliferation and adhesion.

The recruitment of BMSCs in vitro by HCLS was ascertained by incubating a whole rabbit cranial bone marrow cell suspension for 48 h (Fig. 5a1, a2, Supplementary Fig. 24a). The scaffolds were rinsed thoroughly to remove nonadherent cells. Next, the anchored BMSCs were stained with F-actin and cell surface glycoprotein (CD44), which was used as a characteristic marker to identify the phenotype of stem cells. The intensity of CD44 staining and cell morphology suggested the optimal recruitment ability of stem cells in HCLS, and a significantly higher spreading area of adhered stem cells. Subsequently, a transwell-migration assay was conducted to investigate the migration-inducing effects of scaffolds on BMSCs (Fig. 5a3, a4, Supplementary Fig. 24b). Compared with other hydrogels, HCLS could mobilize more directional migration and invasion of BMSCs.

Next, the in vivo ESC recruitment by HCLS was evaluated in a rabbit cranial defect model ($\Phi = 9$ mm). After one week of implantation, the accumulative ESCs were identified by CD44 IF staining, except for the Col I group, which had been completely degraded (Supplementary Fig. 25a). The cytoskeletal morphology in HCLS revealed evenly distributed fusiform shape cells relative to attached round cells in the other two groups, and the SEM images further confirmed the cell morphology (Fig. 5b, c). The CLSM images presented highly elongated actin filaments (red) and CD44 positive expression (green) surrounding the cell nuclei (blue), which confirmed the obvious attachment growth of ESCs in HCLS (Fig. 5d). The dual turntable confocal scanning images of CD44 staining and quantitative analysis clearly revealed that more cells adhered and grew into HCLS with a total number of 38900 cells, in comparison to 30650 cells and 20470 cells in DCLS and HAD, respectively (Fig. 5e1–3, Supplementary Fig. 25b). The

ratio of ESCs with positive CD44 expression in HCLS was 66.1%, which was significantly higher than that in DCLS (34.2%) and HAD (31.9%) (Fig. 5e4). Although other cell types, including preosteoblasts, pericyte-like cells, and endothelial cells, might also express CD44, ESCs with high expression levels of CD44 are primarily contributor cells during wound healing of natural bone and bone regeneration[46,47]. Accompanied by enhanced staining of osteoinductive markers (Runx2 and bone morphogenetic protein 2 (BMP-2)), we, therefore, speculated that the CD44 positive cells in the scaffolds were mainly ESCs (Fig. 5f1–3, Supplementary Fig. 25c). Interestingly, the number of CD44 positive cells on the edges of the HCLS was significantly greater than that in the middle, suggesting that ESCs migrated mainly from the edge to the middle after implantation. To confirm that the recruited cells were BMSCs, CD90 immunofluorescence staining was further performed. As shown in Supplementary Fig. 26, more obvious positive expression of the CD90 marker could be found in HCLS than in DCLS. In addition, the number of CD90-positive cells on the edges of the HCLS was significantly greater than that in the middle.

We further investigated scaffold-mediated angiogenesis and osteogenesis in vivo for 1 week by IF and H&E staining (Fig. 5g1±3, Supplementary Fig. 25d). Blood vessels (green area) were identified in all groups; these newborn blood vessels infiltrated into the HCLS broader area and presented a notably higher density ($1.97 \pm 0.54$ mm/mm²) than in other groups. The enhanced fluorescence signal in HCLS implied stronger osteogenic differentiation. Newly formed bone and vessels were found in HCLS by H&E staining (Fig. 5h), which was consistent with the ex vivo osteogenesis results in the nude mouse subcutaneous model.

**The osteogenic differentiation mechanism of HCLS by transcriptomic analysis.** Transcriptomic analysis of BMSCs cultured on HCLS, DCLS, and Col I was applied to explore the underlying mechanism of osteogenic differentiation. The Pearson correlation and principal component analysis were used to assess the specimen's stability. The correlation coefficients of samples in each group were within acceptable ranges ($R^2 > 0.94$, $N = 3$), indicating good biological repetition, while the value of samples among different groups also illustrated obvious gene expression differences in HCLS vs. Col I and DCLS vs. Col I ($R^2 < 0.85$, $N = 3$), as well as potential similarity in HCLS vs. DCLS ($R^2 > 0.94$, $N = 3$) (Fig. 6a1, 2). There were many gene expression differences (1502 genes in HCLS vs. DCLS, 2240 genes in HCLS vs. Col I, and 2378 genes in DCLS vs. Col I), and they shared 12427 genes (Fig. 6a3). The number of differentially expressed genes between HCLS vs. Col I and DCLS vs. Col I was not significant. Thus, the HCLS vs. DCLS and DCLS vs. Col I groups was used as pairwise comparisons to analyze the differences caused by the introduction of HAD and μHAp. As shown in Fig. 6b1–4, c1–4, the volcano plots and heatmaps showed that 366 upregulated and 133 downregulated genes (DCLS vs. Col I), 58 upregulated and 29 downregulated genes (HCLS vs. DCLS) were involved in the top enriched up-GO (Gene Ontology) terms. In addition, 207 upregulated and 70 downregulated genes (DCLS vs. Col I), 19 upregulated and 5 downregulated genes (HCLS vs. DCLS) were found in the top enriched up-KEGG (Kyoto Encyclopedia of Genes and Genomes) pathways.

The differentially expressed genes were collected and divided into four typical categories, including cellular activity, osteogenesis, angiogenesis, and ESC recruitment. Significant differences in each category could be found in the volcano plots and heatmaps in the DCLS vs. Col I (Supplementary Fig. 27a1–4, b1–4) and HCLS vs. DCLS groups (Supplementary Fig. 28a1–4, b1–4), while their differentially expressed genes for each category were also

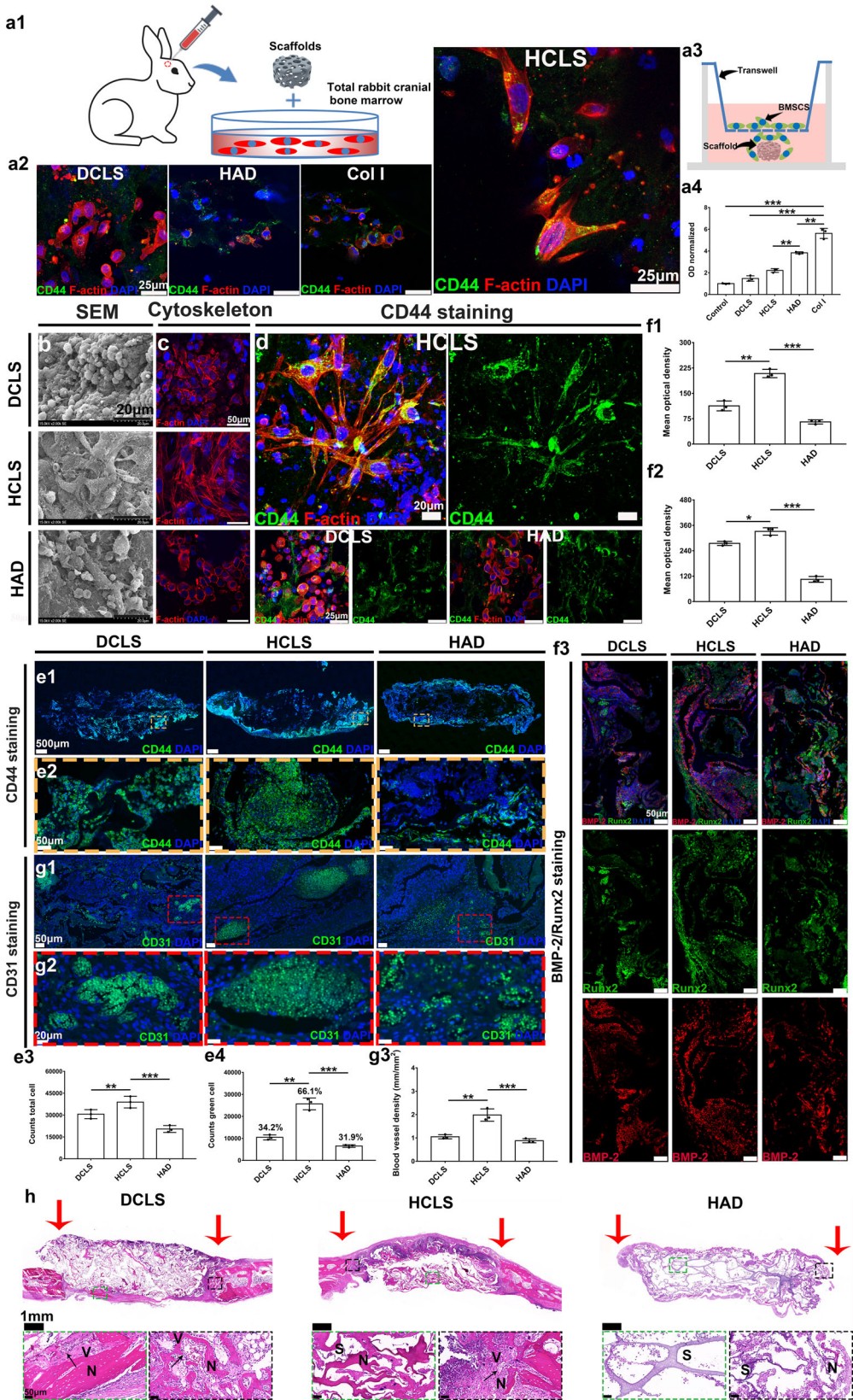

respectively collected to perform GO database analysis (Supplementary Fig. 27c1–4 and Fig. 28c1–4). Generally, the upregulated genes in DCLS vs. Col I (Fig. 6d1) were enriched orderly in ECM, collagen binding, chemotaxis, signaling receptor activity, cell migration, cell adhesion, angiogenesis, and ossification. Compared with DCLS, the HCLS (Fig. 6d2) further enhanced the

related gene expression in ECM structure and organization, integrin binding, epidermal cell differentiation, ossification, peptide binding, vasculogenesis, and labyrinthine layer blood vessel development.

The relevant top enriched up-KEGG pathways speculated that HAD in DCLS (Fig. 6d3) could not only increase cell migration

**Fig. 5 Endogenous stem cells recruitment by HCLS. a1** Schematic depiction of in vitro BMSCs recruitment by incubating various scaffolds in a total rabbit cranial bone marrow for 48 h. **a2** CD44 immunofluorescent staining imaged by CLSM. **a3** Schematic depiction of transwell assay in studying BMSCs migration to various scaffolds. **a4** Quantification of migrated BMSCs by dissolving crystal violet and spectrophotometrically measured at 573 nm, the optical density (OD) resulted was normalized by control. BMSCs alone was used as control (**p = 0.0028$, ***p = 0.0001$, ***p = 0.0002$, ***p = 6.7070 \times 10^{-5}$). $n = 3$ biologically independent replicates. **b** SEM images of endogenous cells on the surface of various scaffolds. **c** CLSM images (rhodamine-phalloidin/DAPI staining) of endogenous cells on the surface of various scaffolds. **d** CLSM images of CD44/F-actin/DAPI staining of various scaffolds. **e1** Confocal quantitative images (CD44/DAPI staining) of ESCs in scaffolds. **e2** High-magnification images (CD44/DAPI staining) of ESCs in scaffolds. **e3** Quantitative analysis of the total number of cells in scaffolds (*p = 0.0470$, **p = 0.0023$). **e4** Quantitative analysis of the number of ESCs in scaffolds (***p = 0.0008$, ***p = 0.0003$). $n = 3$ biologically independent replicates. **f1–3** BMP-2 and Runx2 immunofluorescence staining at one week of different treatments and semiquantitative analysis (**f1**: ***p = 0.0009$, ***p = 5.5298 \times 10^{-5}$; **f2**: **p = 0.0078$, ***p = 5.8847 \times 10^{-5}$). **g1** CLSM images of CD31 immunofluorescence staining. **g2** High-magnification images (CD31/DAPI staining) in scaffolds. **g3** Quantitative analysis of the distribution of blood vessels inside various scaffolds (**p = 0.0044$, **p = 0.0023$). $n = 3$ cells examined three independent experiments. **h** H&E staining of different scaffolds after one week (V: new vessels. N: new bone tissue. S: scaffolds). (Two-sided comparison, error bars represent standard deviation, *p < 0.05$, **p < 0.01$, and ***p < 0.001$).

and adhesion (cytokine-cytokine receptor interaction, chemokine signaling pathways, cell adhesion molecules, and focal adhesion), but also improve proinflammatory expression and M2 macrophage polarization (NF-κB and peroxisome proliferator-activated receptor (PPAR) signaling pathways)[48]. In addition, DCLS significantly promoted stem cell development and matrix formation (signaling pathways regulating pluripotency of stem cell, ECM-receptor interaction, and MAPK signaling pathway), and subsequently enhanced angiogenesis (hematopoietic cell lineage) and osteogenesis (PI3K-Akt signaling pathway, HIF-1 signaling pathways, and TGF-β signaling pathways) in relative to Col I. Furthermore, HCLS that contained μHAp particles accelerated osteogenesis (highest enrichment in the PI3K-Akt signaling pathway)[49] by promoting cell adhesion (focal adhesion, mucin-type O-glycan biosynthesis, and Rap 1 signaling pathway) and activity (cell cycle, ECM-receptor interaction)[50], as well as potential inflammatory regulation (phospholipase D signaling pathway, enhancing the transcriptional activity of NF-κB, and promoting the transcription and expression of cyclin D1 and vascular endogenesis factor) (Fig. 6d4)[51]. Specific gene heatmaps and their corresponding protein-protein interaction networks clarified the process, confirming that the introduction of HAD and μHAp might play a positive role in recruiting ESCs by activating the expression and interaction of osteogenic/angiogenic related proteins (Fig. 6e1–4). The transcriptomic data analysis was consistent with previous experimental results in Figs. 3–5, which indicated that by integrating HAD and μHAp, HCLS might accelerate integrin/peptide/factor binding to recruit ESCs, and then perfect the ECM structure and organization, promote vasculogenesis and promote more intense and timely osteogenesis.

**Skull reconstruction by HCLS in a rabbit model.** Both DCLS and HCLS precisely matched bone defects and had been in close contact with the host bone tissue after 4 weeks of implantation in New Zealand white rabbits (Supplementary Fig. 29a, Φ = 9 mm), while large amounts of new bone tissue were well-integrated with the surrounding tissue after 12 weeks (Supplementary Fig. 29b). The SEM images of cross-sections at defect sites showed that the thickness of new bone tissue changed from 0.9 mm to 1.5 mm in HCLS, while there was no obvious change in other groups (Supplementary Fig. 30a). The dense and smooth interface in HCLS suggested good osteointegration ability (Supplementary Fig. 30b). The push-out test was then performed to evaluate the interfacial binding force between the host and new bone (Supplementary Fig. 30c). The breaking load by the push-out test increased from 36.7 N to 68.5 N in the HCLS group after 12 weeks of implantation, which was close to normal (80.5 N).

Coronal images of micro-CT analysis presented new bone formation around the edges of defects (Fig. 7a, b), which had already expanded to the center of the bone cavity in HCLS at 4 weeks and formed a more complete bone structure at 12 weeks. However, visible and obvious defects were still present in the DCLS group. In the blank group, only a small amount of new bone formed at the defect edge. The radiographic images were consistent with the micro-CT results. At 4 weeks, there was limited bone formation in the blank group (18.7 ± 3.5%, bone volume/tissue volume (BV/TV)), while new bone tissues had infiltrated into the bone cavity in the DCLS (23.5 ± 3.7%, BV/TV) and HCLS (47.3 ± 4.4%, BV/TV) groups (Fig. 7c1). After 12 weeks, the relative bone volume fraction (BV/TV) of the HCLS group increased to 79.6 ± 7.3% in comparison to 64 ± 5.7% and 19.1 ± 2.1% for the DCLS and blank groups, respectively. Additionally, critical factors for osteogenesis, including the bone mineral density (238 mg HA/cm³), bone cover area (97%), trabecula thickness (0.28 mm), and trabecula number (1.75 mm$^{-1}$) in the HCLS groups showed the highest value after 12 weeks of implantation (Fig. 7c2–5), and the decreased trabecular spacing (0.37 mm) implied a more complete bone structure (Fig. 7c6).

Osteogenesis and angiogenesis-related gene expression, including Col I, Runx2, BMP-2, OCN, osteopontin (OPN), and VEGF, was conducted at 12 weeks by quantitative real-time polymerase chain reaction (Q-PCR) (Fig. 7d1–6). Generally, there are three stages in the osteodifferentiation of MSCs: proliferation, matrix maturation, and matrix mineralization[52]. The early differentiation stage involves increased Col I matrix and Runx2 and BMP-2 secretion, and the later stage is characterized by high levels of extracellular matrix proteins, OCN and OPN, which are observed during osteoblast mineralization[53–55]. The HCLS group showed significantly higher expression of Col I, Runx2, and BMP-2 than the other groups after 12 weeks of implantation. Meanwhile, the expression of OCN, OPN, and VEGF was also the highest in HCLS, suggesting significant matrix mineralization and blood vessel formation. In addition, WB of key osteogenic proteins (Runx2 and BMP-2) also presented the highest expression in HCLS, which was consistent with the Q-PCR results (Fig. 7d). The Q-PCR and WB detection results of VEGF (Fig. 7d6, e4) indicated that HCLS exhibited higher gene expression levels than DCLS, but slightly lower protein expression.

Next, regenerative sections were studied by H&E, Masson's trichrome (MT), and CD31 IF staining (Fig. 8a1–3, Supplementary Fig. 31a1–3 and b). H&E staining showed that only a small amount of newly formed bone matrix appeared in the defect boundary in the blank group, even after 12 weeks. A large amount of mature dense bone tissue completely occupied the defect area and integrated tightly with the host bone in the treatment groups at 12 weeks. Newborn bone lacunas and central canals were observed in HCLS,

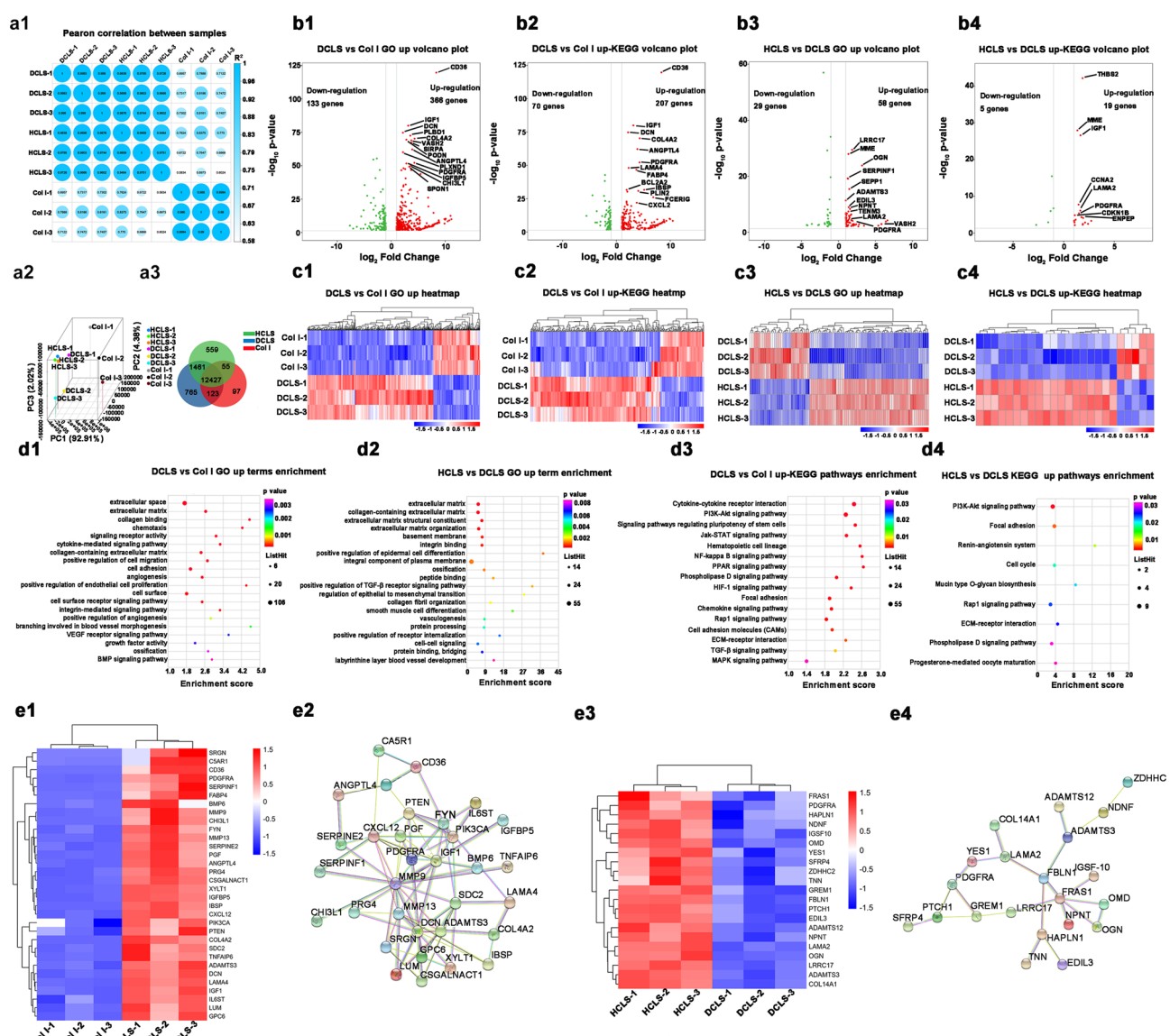

**Fig. 6 Self-adhesive and flexible scaffold regulates gene expressions related to osteogenesis, angiogenesis, and ESCs recruitment in vitro. a1** Heatmap of Pearson correlation between samples. **a2** 3D image of principal component analysis of different samples. **a3** Venn diagram of the number of differentially expressed genes in different samples. **b1–4** Volcano plot of transcriptomic analysis of differentially expressed genes. $n = 3$ independent experiments per group. **c1, c3** Heatmap analysis of differentially expressed genes involved in top enriched up-Gene Ontology (GO) database. **c2, c4** Heatmap analysis of differentially expressed genes involved in top enriched up- Kyoto Encyclopedia of Genes and Genomes (KEGG) pathways. **d1, d2** Enriched GO up terms of DCLS versus Col I and HCLS versus DCLS. **d3, d4** Enriched KEGG pathways of DCLS versus Col I and HCLS versus DCLS. **e1, e3** Heatmap of differentially expressed genes involved in cellular activity, osteogenesis, angiogenesis, and ESCs recruitment of DCLS versus Col I and HCLS versus DCLS. **e2, e4** String interaction network of differentially expressed genes involved in cellular activity, osteogenesis, angiogenesis, and ESCs recruitment of DCLS versus Col I and HCLS versus DCLS. (Two-sided comparison, Error bars represent standard deviation, *$p < 0.05$, $n = 3$ biologically independent replicates).

which presented a relatively complete laminar structure similar to the normal skull (Fig. 8a1 and Supplementary Fig. 31a1). There was no obvious formation of collagen fibre in the blank group by MT staining (Fig. 8a2 and Supplementary Fig. 31a2), but many immature collagens (blue regions) were observed in DCLS. The largest area of robust mature collagen (red regions) in HCLS after 12 weeks indicated impressive osteogenic ability. Scaffolds-mediated host cell angiogenic differentiation in vivo was confirmed by positive staining of CD31(Fig. 8a3 and Supplementary Fig. 31a3). Host vascular endothelial cells (CD31$^+$) infiltrated and formed blood vessels inside the DCLS and HCLS at postoperative weeks 4 and 12, especially in the HCLS, which was well developed with continuous endothelial lining cells. These results confirmed that HCLS facilitated the ingrowth of host blood vessels and accelerated skull

regeneration. In addition, immunohistochemical staining (BMP-2, Runx2, OCN, and VEGF) was performed after 4 and 12 weeks after implantation to investigate possible osteogenic-related expression during the bone regeneration process (Fig. 8b1–4, c and Supplementary Fig. 31c1–4, d). Higher osteo-related expression of BMP-2, Runx2, and OCN in the DCLS and HCLS groups was confirmed. The higher positive expression level of VEGF in HCLS demonstrated a greater potential for facilitating vascularization[56,57].

**Skull reconstruction by HCLS in a beagle dog model.** HCLS was implanted into the cranial bone defects (Φ = 15 mm) of beagle dogs with intact meninges (Supplementary Fig. 32a, b). The radiographic images of HCLS revealed that new bone tissue

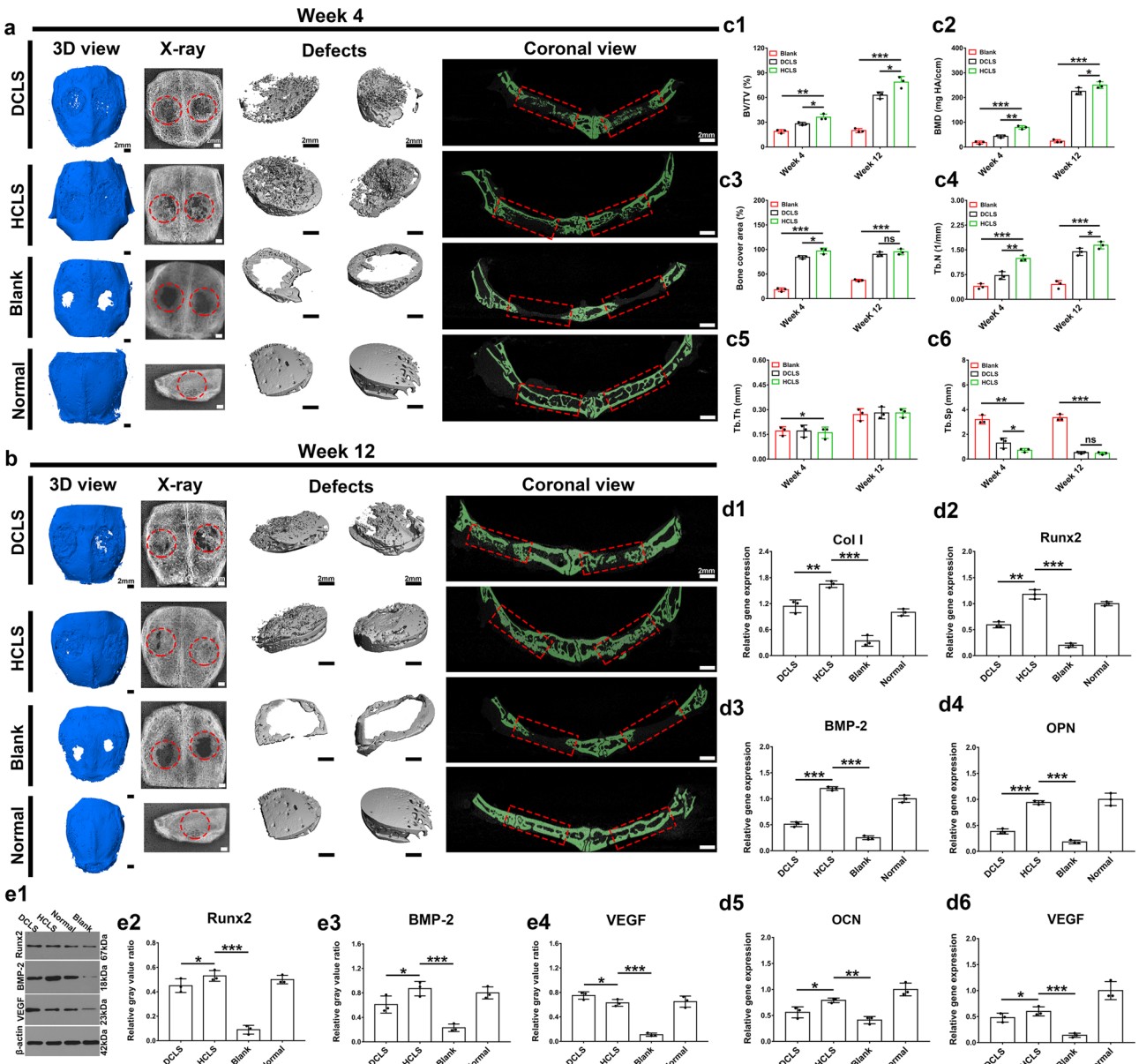

**Fig. 7 In situ skull regeneration enhanced by DCLS and HCLS in rabbit cranial defect model. a, b** Representative micro-CT and X-ray images at 4 and 12 weeks. **c1–6** Quantitative analyses of micro-CT results at 4 and 12 weeks. (all the free-cells scaffolds implantation. Blank: defect alone with no treatment) (**c1**: *$p = 0.0459$, *$p = 0.0324$, **$p = 0.0011$, ***$p = 1.4129 \times 10^{-7}$; **c2**: *$p = 0.0233$, **$p = 0.0019$, ***$p = 2.4899 \times 10^{-5}$, ***$p = 9.8579 \times 10^{-9}$; **c3**: *$p = 0.0141$, ns $= 0.2583$, ***$p = 2.8782 \times 10^{-8}$, ***$p = 3.1798 \times 10^{-7}$; **c4**: ***$p = 0.0003$, *$p = 0.0451$, ***$p = 6.4603 \times 10^{-7}$, **$p = 0.0049$; **c5**: *$p = 0.0429$; **c6**: *$p = 0.01457$, ns $= 0.7563$, **$p = 0.0011$, ***$p = 3.6545 \times 10^{-7}$). **d1–6** Q-PCR analyses of osteo-related genes, including Col I, Runx2, VEGF, BMP-2, OCN, and OPN in different scaffolds at week 12 (**d1**: **$p = 0.0060$, ***$p = 9.5355 \times 10^{-5}$; **d2**: ***$p = 0.0007$, **$p = 0.0065$; **d3**: ***$p = 3.2801 \times 10^{-5}$, ***$p = 5.2939 \times 10^{-6}$; **d4**: ***$p = 9.4228 \times 10^{-5}$, ***$p = 1.2369 \times 10^{-5}$; **d5**: *$p = 0.0253$, **$p = 0.0012$; **d6**: *$p = 0.0169$, ***$p = 1.4583 \times 10^{-6}$). **e1** Western blot analyses of Runx2, BMP-2 and VEGF expression in different scaffolds at week 12. Uncropped blots in Source Data. **e2–4** Relative content (Gray value ratio) of Runx2, BMP-2 and VEGF protein in different scaffolds at week 12 by WB analysis (**e2**: *$p = 0.0410$, ***$p = 0.0002$; **e3**: *$p = 0.0332$, ***$p = 0.0011$; **e4**: *$p = 0.0453$, ***$p = 0.0006$). (Two-sided comparison, error bars represent standard deviation, *$p < 0.05$, **$p < 0.01$, and ***$p < 0.001$).

integrated tightly with the host bone (Fig. 9a). The representative 3D images by micro-CT suggested that no obvious newly formed bone could be found in the blank group at 12 weeks post-implantation. Even the bone mass reduction was observed, which might be attributed to bone resorption of local osteonecrosis derived from thermal damage by a high-speed ball milling drill during surgery[58–61]. However, a gradual increase in newborn bone tissue was found in the HCLS group. The quantitative analysis further confirmed that the HCLS group had a significantly higher

BV/TV value ($35.7 \pm 5.1\%$, 4 w; $58.7 \pm 7.6\%$, 12 w), new bone volume ($126.1 \pm 15 \text{ mm}^3$, 4 w; $207.1 \pm 27.3 \text{ mm}^3$, 12 w) and new bone cover ratio ($71.7 \pm 6.9\%$, 12 w), demonstrating that HCLS promoted cranial bone regeneration in beagle dogs (Fig. 9b1–4).

H&E staining in Fig. 9c showed that only muscle and fibrous tissues appeared in the defect areas even after 12 weeks in the blank group. However, obvious newly formed bone and vessels were observed in the HCLS group at 4 weeks, which further expanded to the central region of the defects after 12 weeks. Moreover, MT

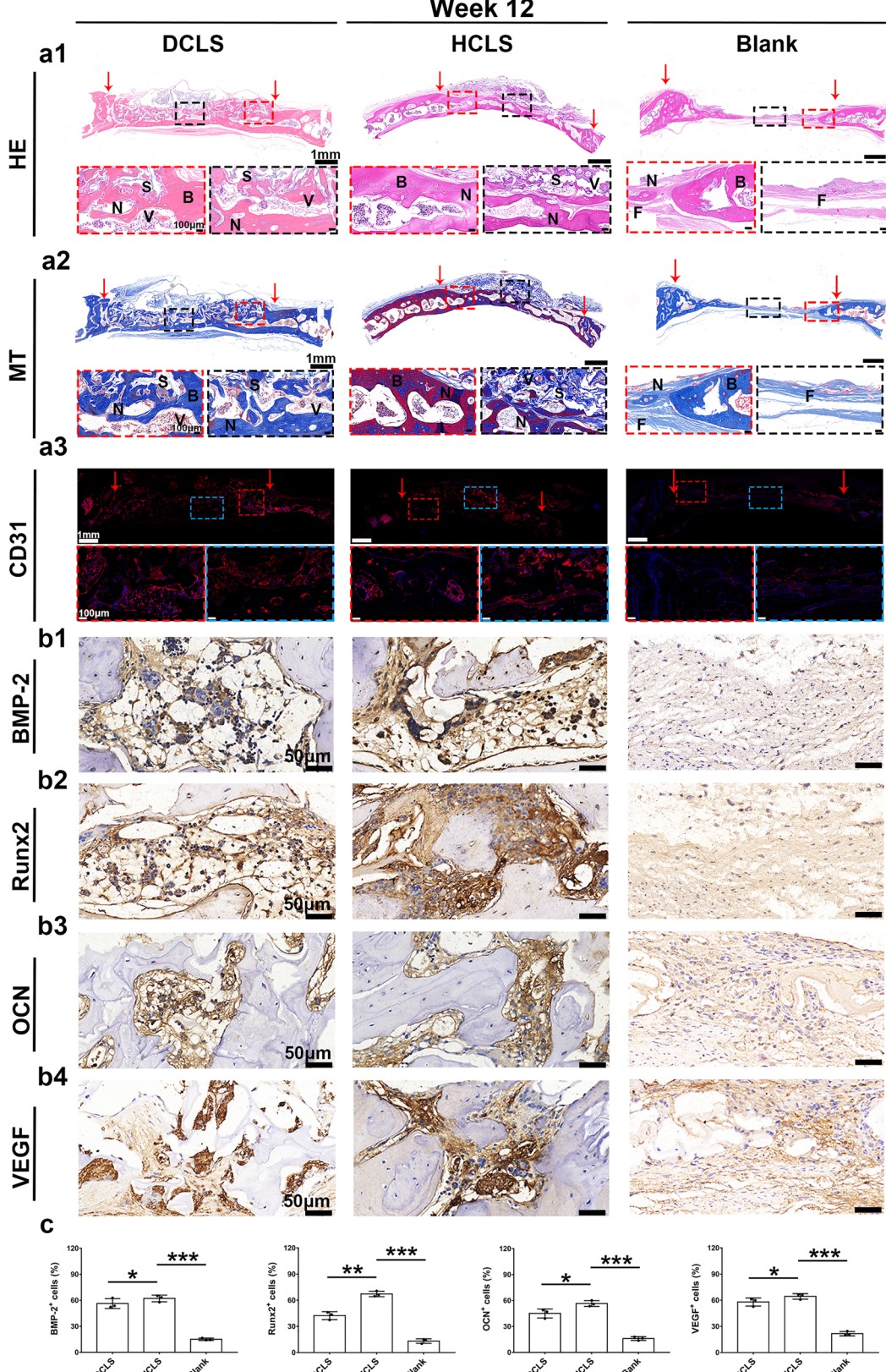

staining (Fig. 9d) was used to evaluate the formation and maturation of bone tissue. The loose connective tissues and mussel tissues filled the defect sites of the blank group. In contrast, in the HCLS, many immature collagens (blue regions) were observed at 4 weeks, and the area of robust mature collagen (red regions) was further expanded after 12 weeks. These results were in accordance

with the micro-CT data. Potential angiogenesis was also assessed by IF staining of CD31 (green). Significant positive expression was observed in HCLS, and semiquantitative analysis results confirmed the obviously enhanced fluorescence with the time extension, suggesting progressively denser neovascularization, which was conducive to new bone formation (Fig. 9e, f).

**Fig. 8 Representative staining images of regenerated tissue at week 12. a1** H&E staining of regenerated bones induced by different scaffolds at week 12 after operation. **a2** Masson's trichrome staining at week 12 after the operation. **a3** CD31 immunofluorescence staining of regenerated vessels at week 12. (Row 1: Overall observation of the cranial defect repair. The red arrow indicates the initial boundary of the defect. Row 2: Magnified view of the center and boundary site of the defects). (N: new bone tissue. S: scaffolds. V: new blood vessels (black arrow). F: fibrous tissue. B: old bone boundary). **b1–4** Representative immunohistochemistry images of BMP-2, Runx2, OCN, VEGF at week 12 after implantation. **c** Quantitative analysis of positive cells at week 12. (all scaffolds without cells. Blank: defect alone with no treatment) (***$p = 3.7616 \times 10^{-5}$, *$p = 0.0114$; ***$p = 2.1035 \times 10^{-5}$, **$p = 0.0032$, ***$p = 6.41247 \times 10^{-5}$, *$p = 0.0355$, ***$p = 5.2544 \times 10^{-5}$, *$p = 0.0409$). (Two-sided comparison, error bars represent standard deviation, *$p < 0.05$, **$p < 0.01$, and ***$p < 0.001$).

## Discussion

We developed an instantly fixable and self-adaptive scaffold (HCLS) that mimics the composition and structure of cranial bone matrix to promote host vascularized osteogenesis. The unique feature of the HCLS was the hybrid cross-linked bonding among Col I nanofibrous surfaces, μHAp particles, and HAD hydrogel networks. This bonding provided an effective approach to improve the structural stability and tissue adhesion ability of the scaffold with relatively higher porosity and interconnected porous structure, which contributed to cellular infiltration, vascularization, and tissue remodeling. Satisfactory interface integration was achieved by the HCLS after 7 days of implantation in vivo and ensured the invasion, infiltration, and migration of ESCs into the scaffold.

Currently, patients normally have to wait for 3–6 months after craniectomy to relieve intracranial pressure and then undergo cranioplasty[4]. In consideration of the clinical application of HCLS, the self-adhesion and flexibility characteristics of HCLS might not only maintain the integrity of the scaffold through deformation and adhesion but also adapt to the changes in intracranial pressure, so it would allow immediate cranial defect restoration in a single-stage procedure combining craniectomy and cranioplasty. Mechanical tests demonstrated a good mechanical match between HCLS and the bony host environment. It could maintain structural integrity through flexible deformation to a certain extent when stimulated by external stress in the defect. Even with blood infiltration, HCLS could still maintain a well flexible property, suggesting great potential for clinical applications. The HCLS provided cranial defect restoration in an instantly fixable form without the use of an auxiliary fixation device and exogenous growth factors/cells, thus enhancing its clinical translatability. In addition, being implanted with biomaterials with such high flexibility often avoids inflammation owing to friction with brain tissue. Our results from the rabbit and beagle dog models demonstrated the cases for immediate repair after cranial tissue removal.

Implantable biomaterials with favorable immune responses can guide successful osteogenesis and angiogenesis[40,62]. Thus, the in vitro and in vivo immune responses of HCLS and DCLS were evaluated. The proper immune microenvironment provided by HCLS not only facilitated abundant filopodia formation of macrophages in vitro, but also obviously regulated M2 polarization of macrophages with an ascending secretion in endogenous VEGF and BMP-2, which contributed to new vessel formation and bone regeneration in vivo. Transcriptomic analysis results confirmed that the proper immune microenvironment provided by HCLS was attributed to the upregulation of NF-κB and peroxisome proliferator-activated receptor (PPAR) signaling pathways, which was consistent with previous reports that CD206$^+$ macrophages were a rich source of VEGF production[36].

The results of 2D/3D culture in vitro proved that HCLS could create an interactively biomimetic matrix microenvironment, which significantly promoted BMSC proliferation and osteodifferentiation, and accelerated angiogenesis-related factor expression. The structural stability of the scaffolds in a wet environment after implantation in vivo played a vital role in maintaining the defined structures and biological functions. In a well-established nude mouse subcutaneous implantation model, the application of BMSC-laden HCLS confirmed the structural stability, significant ectopic mineralization, and angiogenesis, as well as highly consistent osteoid matrix formation. The HCLS might reconstruct a matching interactive microenvironment with BMSCs in the humoral environment, which facilitated the aggregation, proliferation, and osteogenic differentiation of BMSCs, as well as accompanied angiogenesis. The mechanical strength of HCLS was markedly enhanced through rapid and large amounts of new bone tissue infiltration, thereby providing protection for brain tissue.

Regenerating extensive skull tissue defects by cell-free implants alone has been challenging because spontaneous endogenous cell infiltration to reconstruct cranial tissue is slow. Previous studies have focused on the use of implants combined with exogenous factors or cellular cues for skull reconstruction[63–65]. Although these strategies have achieved some advances in experiments, the burdensome clinical approval process and tumorigenic risk of exogenous stem cells and factor products have postponed their practical application. This work highlights skull remodeling and regeneration by accelerating osteodifferentiation to activate the self-repair function. The in vitro transwell-migration results indicated that the HCLS could accelerate the directional migration of BMSCs into HCLS, and the scaffold could also promote the accumulation of stem cells (CD44 and CD90 markers) derived from whole rabbit cranial bone marrow cell suspension. The in vivo cell recruitment results confirmed the optimal capacity of HCLS for rapidly initiating angiogenesis and osteogenesis in rabbit cranial defects by in-situ recruitment of ESCs and subsequent acceleration of osteodifferentiation in an interactive microenvironment.

Transcriptomic analysis results in vitro verified that the introduction of HAD in DCLS significantly increased gene transcription levels on related protein/chemokine activity, cell migration and adhesion, and M2 macrophage activation. Furthermore, the μHAp particles in HCLS obviously promoted potential osteogenesis and angiogenesis (PI3K-Akt signaling pathway and renin-angiotensin system), as well as specific protein adhesion and cellular activity (focal adhesion and cell cycle). Hence, we inferred that the extracellular matrix-related protein of ESCs could be adsorbed to HCLS by catechol groups and formed cell-adhesive protein domains, thereby allowing cell attachment by binding to integrin receptors on the cell membrane[66]. The research pointed out that implants with specific structures could enrich chemokines, such as *CXCL12-CXCR4*, for recruiting ESCs into the scaffold[67]. Transcriptomic analysis of HCLS confirmed the upregulation of *CXCL12, LAMA4, LAMA2,* and *SRGN* genes (Fig. 6e1, e3), which play a vital role in recruiting monocytes/macrophages and ESCs. It was also found that blood vessels infiltrated into the interior zone of the HCLS, which presented a notably higher vascular density one week after implantation. The angiogenesis response (*PGF, IGF1, PDGFRA,* and *ADAMTS3 genes* in Fig. 6e1, e3) was probably attributed to the synergistic

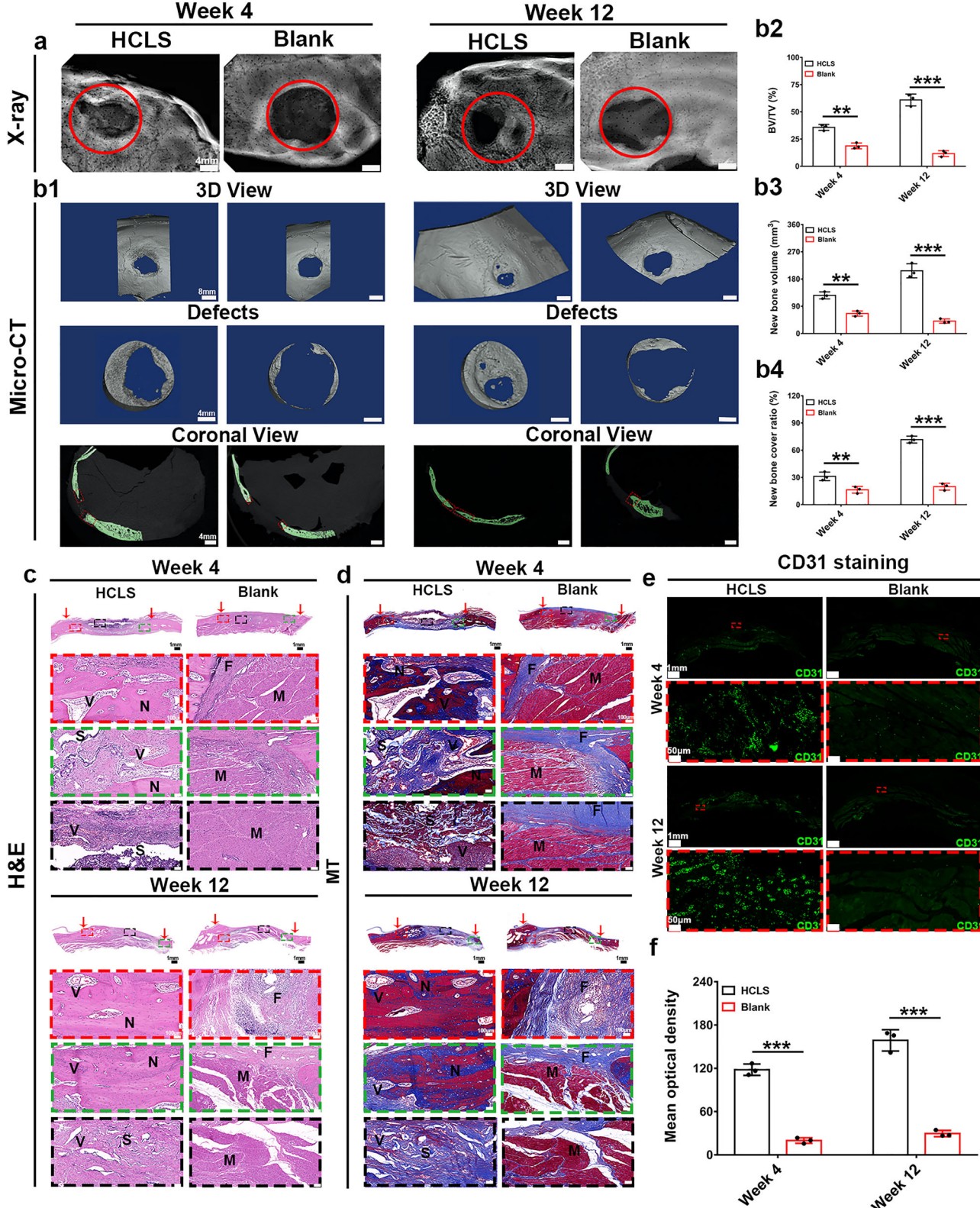

effects of high porosity and structural stability of the scaffold as well as the biological function of recruited ESCs that produced Runx2 and VEGF to potentially modulate osteogenesis and angiogenesis[68]. In addition, the hybrid cross-linked network could provide stable mechanical support for a certain period of time to guarantee structural integrity, endowing vascular development and maturity across the HCLS. In contrast, the relatively fast degradation of DCLS limited the structural integrity, leading to incomplete vascularization. The gene expression of the PI3K-Akt signaling pathway and focal adhesion was prominently upregulated in HCLS, suggesting that the degradation of μHAp and chelating of calcium ions by catecholamine moieties might promote osteogenesis (*OMD, OGN, GREM1,* and *COL14A1* genes in Fig. 6e3). Our results confirmed that the hybrid cross-linked

**Fig. 9 Skull reconstruction by HCLS activating angiogenesis and osteogenesis in beagle dog model. a** X-ray images of the regenerated bone tissue. **b1** Representative micro-CT images and **b2–4** quantitative histomorphometry analyses of bone regeneration in cranial defects in dogs at 4 and 12 weeks (b2: **$p = 0.0074$, ***$p = 1.5931 \times 10^{-7}$; **b3**: **$p = 0.0008$; ***$p = 5.4303 \times 10^{-7}$; **b4**: **$p = 0.0019$, ***$p = 2.51131 \times 10^{-7}$). $n = 3$ biologically independent replicates. **c** H&E staining of regenerated bones at 4 and 12 weeks after operation. **d** Masson's trichrome staining of regenerated bones at 4 and 12 weeks after the operation. (Row 1: overall observation of the cranial defect repair. The red arrow indicates the initial boundary of the defect. Row 2: magnified view of the center and boundary site of the defect). (N: new bone tissue. S: scaffolds. V: new blood vessels. F: fibrous tissue. M: Muscle tissue). **e** Representative IF staining images of CD31 at 4 and 12 weeks after implantation. **f** Semiquantitative analysis of CD31 IF staining at 4 and 12 weeks. (all scaffolds without cells. Blank: defect alone with no treatment) (***$p = 8.1804 \times 10^{-7}$, ***$p = 9.7494 \times 10^{-8}$). (two-sided comparison, error bars represent standard deviation, *$p < 0.05$, **$p < 0.01$, and ***$p < 0.001$).

network was required for realizing ESC recruitment and angiogenesis, as well as subsequent skull regeneration and remodeling.

The long-term implantation (12 weeks) of HCLS in rabbit cranial defects ($\Phi = 9$ mm) confirmed that the skull was successfully reconstructed with a bone mineral density of 238 mg HA/cm$^3$ and a bone cover area of 97%. Histological analysis and immunofluorescence staining verified abundant mineralized bone matrix and vascular formation. The WB and Q-PCR results revealed the coincident expression of genes and proteins related to osteogenesis and angiogenesis, except for VEGF. This could be explained by the fact that gene expression is divided into transcriptional and translational levels, namely, the mRNA and protein levels. The time and location of transcription and translation of eukaryotic gene expression exhibit spacetime intervals. After transcription, posttranscriptional processing, degradation of transcriptional products, translation, posttranslation processing, and modification occur. Thus the transcriptional level and the translation level are not exactly the same[69]. In addition, as a result of a positive feedback mechanism, the high expression of Runx2 could further enhance VEGF secretion, which in turn stimulates an angiogenic response[70]. Although HCLS also presented significant osteogenesis and angiogenesis in the beagle dog model, the therapeutic effect was significantly weaker than that in the rabbit model. According to literature reports[71–73], the thickness of rabbits, beagle dogs, and human skull is ~2 mm, 3 mm, and 10~15 mm, respectively, and the corresponding elastic moduli are 1~2 MPa, 7.5 GPa, and 8~15 GPa, respectively. In addition, rich muscles in the beagle skull enhanced the mechanical environment of the defect area. Herein, the reasons for this difference could be conjectured as follows: The highly developed muscular tissue on the skull of beagle dogs exerted great pressure on the soft implant materials, leading to the structural instability and excessive deformation; in the absence of exogenous fixation, the adhesive force of HCLS might not be enough to support long-term fixation at the defect sites of highly moving beagle dogs. In future studies, the structural design of implants should be further optimized to match the mechanical environment of defect sites.

In conclusion, we have developed an instantly fixable and self-adaptive scaffold by a highly biomimetic hybrid cross-linked construct of a natural skull. The scaffold could integrate with the host tissue by instant adhesion, and respond to external stress in defect sites by flexible deformation. It promoted the recruitment of immune cells and stem cells in vitro and in vivo, regulated macrophage M2 polarization, and initiated rapid angiogenesis and osteogenesis, thus laying a reliable foundation for cranial defect restoration in rabbit ($\Phi = 9$ mm) and beagle dogs ($\Phi = 15$ mm) models. This strategy might represent a widely considered cell/factor-free regenerative strategy, which is expected to provide a meaningful clinical alternative for cranial regeneration.

## Methods

**Preparation of dopamine-modified HAD**. HAD was synthesized by conjugating dopamine (62-31-7, Best-reagent Corporation, China) to hyaluronic acid (HA,

9067-32-7, Bloomage Freda Biopharm Corporation, China, MW = 0.3 MDa) using 1-ethyl-3-(3-(dimethylamino)-propyl carbodiimide (EDC, 25952-53-8, Best-reagent Corporation, China) and *N*-hydroxysulfosuccinimide (NHS, 6066-82-6, Best-reagent Corporation, China). Briefly, 400 mg of HA was dissolved in 35 mL of 1× phosphate-buffered saline (PBS). EDC (766 mg) and NHS (230 mg) were slowly added to a mixed solution with the molar ratio of HA/EDC/NHS at 1:4:2. After 30 min stirring, 1 mmol of dopamine was added to the mixture and the pH value was continuously monitored and maintained at 5 for 12 h. The entire reaction was carried out under anaerobic conditions. After the reaction, the solution was purified by dialysis under acidic conditions for 3 days and was subsequently lyophilized to get a white powder.

**Fabrication of self-adhesive and flexible scaffold**. In all, 2.5% (w/v, in Milli-Q water) HAD and 2.5% Col I (w/v, in 0.5 M acetic acid solution) (9007-34-5, Trauer company, Guangzhou, China) were mixed and placed on a shaker at 4 °C until they became transparent. Then the solution was adjusted to pH 7.4 and kept for 24 h at 37 °C, after lyophilization, DCLS was obtained. Micron hydroxyapatite (μHAp, 12167-74-7) was provided by the Biomaterial Engineering Research Center of Sichuan University. For the preparation of HCLS, μHAp (1 w/v%) was firstly dispersed in HAD solution (2.5 w/v% in Milli-Q water) under sonication for 30 mins and then mixed with 2.5% Col I (w/v, in 0.5 M acetic acid solution) under ice bath conditions. Then the pH was rapidly adjusted to 7.4 to achieve quick gel formation and to avoid μHAp settling. For experimental control, 2.5% (w/v, in Milli-Q water) HAD and 2.5% Col I (w/v, in 0.5 M acetic acid solution) were adjusted to pH 7.4 with 1 M NaOH (1310-73-2, CHRON CHEMICALS, China) and kept for 15 min at 37 °C to form HAD and Col I hydrogels.

**Characterization of scaffold**. Hydrogels were imaged in a low vacuum using scanning electron microscopy with X-ray microanalysis (SEM/EDS, S-4800, Hitachi, Japan) to characterize the microstructure and chemical compositions. Masson's trichrome and calcein (10 μM for 20 min) staining were used to analyze the distribution of Col I and μHAp in HCLS. The crystalline phase of μHAp was assayed using XRD (Empyrean, PANalytical B.V, 0.02°/s, 5~80°). A thermogravimetric analyzer (TGA, METTLER TOLEDO, Switzerland) was used to measure the thermogravimetric curve of samples (10~15 mg) under nitrogen gas protection (30~800 °C, 10 K/min). The moisture and heat stability of the samples were characterized by a differential scanning calorimeter (DSC1, METTLER TOLEDO, Switzerland) under nitrogen gas protection (25~160 °C, 10 K/min). The chemical structure of HAD was measured by $^1$H NMR (400 MHz, Bruker AMX-400, USA) using D$_2$O as a solvent and MestReNova Version 10.0.1 and ChemDraw Version 18.0 softwares were used for $^1$H NMR data analysis. The chemical structure of various scaffolds was characterized by FTIR spectroscopy (Nicolet 6700, USA), and the content of catechol in various samples was confirmed by UV spectrophotometer (Hitachi F-7000 Fluorescence Spectrophotometer) with a dopamine standard measuring absorbance at 280 nm.

The swelling property of DCLS, HCLS, HAD, and Col I hydrogels was conducted. Briefly, disc-shaped hydrogels were prepared by immediately injected into molds (diameter 8 mm, height 3.0 mm) and kept for 24 h at 37 °C. Thereafter, hydrogels were immersed in 10 mL PBS buffer (pH 7.4) and placed in a constant temperature shaker (90 rpm at 37 °C, ZHWY-2012C, Shanghai Zhicheng, China). Diameters at ten different positions per sample were measured from gross view images on days 3, 7, and 14 ($D_c$) by Image J software. The size change ratio was calculated by the following formula: Size change ratio = $D_c/D_o \times 100\%$, $D_o = 8$ mm. The storage modulus (G′) of swelled hydrogels ($n = 3$) was determined by DMA (TA-Q800, USA) instrument with a fixed frequency of 5 Hz at constant room temperature. The testing parameters were set as the amplitude of 40 μm, preload force of 0.002 N, and force track of 105%. Additionally, the disintegration performance of DCLS, HCLS, HAD, and Col I hydrogels was tested. Briefly, the prepared disc-shaped hydrogels were washed, freeze-dried, and weighed (W$_0$), and they were immersed in 10 mL PBS containing hyaluronidase (100 U) and type I collagenase (100 U) in a constant temperature shaker (90 rpm at 37°C). After that, samples were taken out, washed in distilled water, freeze-dried, and weighed again (Wr). The disintegration behavior of various samples was expressed as a percentage of weight retention and calculated as follows: Weight remaining percentage = Wr/Wo × 100%. Shear tensile test was performed by Shimadzu

mechanical testing machine (EZ-LX 1 kN) to measure the adhesive strength of DCLS, HCLS, HAD and Col I. Hydrogels were applied to different substrates (glass, polyethylene, titanium, rabbit cranial bone (Chengdu Dossy Experimental Animals CO., LTD, China) and porcine skin (Chengdu Dossy Experimental Animals CO., LTD, China)) with a bonding area of 30 mm × 25 mm. Adhesive test was immediately conducted when the hydrogel was attached to the substrate. Origin 8.0 and Graphpad Prism 7 were used for all statistical analyses. For analyzing calcium ion chelation, DCLS and HCLS were immersed in 0.3 mol/L CaCl$_2$ solutions (DCLS-0.3 M Ca$^{2+}$ and HCLS-0.3 M Ca$^{2+}$) at 37 °C for 12 h, respectively. The mechanism of calcium ion chelation was studied by XPS (Kratos AXIS ULTRA DLD) at a full spectrum power of 10 Kv, 7 mA, and peak spectral power of 10 Kv, 15 mA. C1s binding energy (284.3 eV) was selected as the standard for energy correction. Every sample was measured in three replicates.

**Mechanical characterization of self-adhesive and flexible scaffold.** Samples with or without invasive blood were used for tensile (EZ-LX 1 kN) and compressive tests, respectively. For in vivo mechanical test, 12 adults New Zealand white rabbits (2.5–3.0 kg, male, 2.5–3 months old, 0044063, Chengdu Dossy Experimental Animals Co., LTD., China) were used in this study. After anesthesia, scaffolds were transplanted into a 9 mm-diameter defect on each side of cranial bone ($n = 3$), and animals were grouped as follows: blank (defected only), DCLS, HCLS, and normal (without surgery). Animals were euthanized on days 7 and 14 post-surgery, and the interface between scaffold and host bone tissue was photographed by stereomicroscope (SYCOP 3, ZEISS, Germany) and SEM. Push-out test was conducted to evaluate osseointegration performance at weeks one and two (Shimadzu, EZ-LX 1 kN). Furthermore, the compressive storage modulus (G′) of various implants was determined by DMA instrument (TA-Q800, USA), with a multi-frequency of 1, 2, and 5 Hz. The testing parameters were set as previously mentioned. Every sample was measured in three replicates.

**Response of scaffolds to macrophages in vitro.** Mouse RAW 264.7 macrophages (SCSP-5036, Cell Bank of Chinese Academy of Sciences, Shanghai, China) were maintained in Dulbecco's modified Eagle's medium (DMEM, 11965084, Gibco, USA) supplemented with 10% standard fetal bovine serum (FBS, 10100147, Gibco, USA) and 1% penicillin/streptomycin (CAS: 15140122, Gibco, USA). All samples were sterilized using 75% ethanol (64-17-5, CHRON CHEMICALS, China), and then kept 15 min under ultraviolet light. A 1 mL of cell suspension (RAW 264.7, $5 × 10^4$ cells/mL) was cultured on DCLS and HCLS ($n = 3$) in 24-well plates. After co-culturing for 3 and 5 days, samples were fixed with 2.5% glutaraldehyde (111-30-8, CHRON CHEMICALS, China) for 4 h, and then gradient-dehydrated with 20, 40, 60, 80, and 100% ethanol for 20 min. Next, cell morphology was observed by SEM (S-4800, Hitachi, Japan). Besides, these cells were permeabilized with 0.5% Triton X-100 (HFH10, Invitrogen) for 15 min, and blocked by 1% bovine serum albumin (9048-46-8, Sigma-Aldrich) for 1.5 h. Subsequently, samples were incubated with primary antibodies against CD206 (ab64693, Abcam) and CD197 (ab32527, Abcam) with 1:100 dilution at 4 °C overnight. Next, samples were further incubated with Goat anti-Mouse IgG (H + L) Highly Cross-Adsorbed Secondary Antibody (A16080, Invitrogen), Alexa Fluor Plus 488 (A32723, Invitrogen, 1:100) for 35 min, followed by 3 min of nuclear staining with DAPI (D1306, Invitrogen). At last, immunofluorescence-stained cells were visualized by CLSM.

**The in vivo immune response of scaffold both in mouse intramuscular and rabbit skull defect model.** All animal studies were approved by the Sichuan University Medical Ethics Committee. All animal procedures were performed in accordance with the guidelines for the care and use of laboratory animals at Sichuan University. Twenty BALB/C mice (N000020, 6 weeks, ~20 g, male, GemPharmatech Co., Ltd, China, $n = 20$) and ten adult New Zealand white rabbits (0044063, 2.5–3.0 kg, male, 2.5–3 months old, Chengdu Dossy Experimental Animals Co., LTD., China, $n = 10$) were used in this study. The mice were fed in separated cages in a temperature, humidity-controlled (~25°C, 50–80%) and 12 h light/dark cycle room. After anaesthetized, various samples (Φ = 5 mm) were implanted into the bilateral thigh muscles of each mouse including HCLS and DCLS groups, respectively. Surgeries of rabbits were similar to the previous surgery procedures. Implants ($n = 20$) were retrieved postoperatively to analyze immune cells infiltration on days 7 and 14 including HCLS and DCLS groups. For characterizing morphological change of macrophages by SEM, samples were fixed in 2.5% glutaraldehyde solution for 48 h at 4°C, subsequently dehydrated against a graded series of alcohol and were dried by critical point drying (Leica EM CPD300) for 90 min. Flow cytometry was used to assess recruitment and phenotype of macrophage in vivo. Briefly, to obtain a single-cell suspension of macrophages, the retrieved implants were isolated by trypsinization (R001100, Gibco) for 15 min at 37 °C and washed with phosphate-buffered saline (MFCD00131855, Sigma-aldrich) three times. The resulting cell suspension was filtered (40 μm) and washed using a staining buffer. Subsequently, murine cells were blocked with CD16/CD32 monoclonal antibody (FCR4G8, eBioscience) for 30 min at 4 °C, and then immune cells were stained with Alexa Fluor 700-conjugated F4/80 (MF48005, Invitrogen), APC-conjugated CD197 (12-1971-83, eBioscience), and FITC-conjugated CD206 (PA5-46879, Invitrogen) for 45 min at 4 °C. Appropriate isotypes were used, and ethidium monoazide bromide staining excluded dead cells. At last, cell suspension

was assessed on an LSRFortessaTM X-20 flow cytometer system (BD Biosciences) and analyzed by FlowJo software (Tree Star).

To measure the cytokine of macrophages, the harvested implants were ground with 800 μL of cold PBS and subsequently centrifuged for 20 min at 300 g in 4 °C to obtain the implant supernatant, which was then stored at −80 °C for ELISA assay. The cytokines of implant supernatant were then measured by commercial ELISA kits (Neobioscience): IL-1β (EMC001b), TNF-α (EMC102a), IL-4 (EMC003), IL-10 (EMC005) and IL-1Ra (EMIL1RNX10, Invitrogen), following the manufacturer's guidance. Histological analysis was performed after fixation and decalcification, and then samples were dehydrated and embedded in paraffin. Sections were stained with hematoxylin and eosin (H&E, MFCD00078111, Sigma-aldrich), and the recruited macrophages were microscopically examined. For immunofluorescence, sections were treated with 0.2% Triton X-100 for 30 min, blocked with 5% goat serum at room temperature for 1 h, and incubated overnight at 4 °C with the antibodies F4/80 (Abcam ab6640, 1:100), BMP-2 (Abcam, ab6285, 1:200), VEGF (Abcam, ab52917, 1:250), CD197 (Abcam, ab32527, 1:200), CD206 (Abcam, ab8918, 1:100) and Goat anti-Mouse IgG (H + L) Cross-Adsorbed Secondary Antibody (Jackson, 115-545-003, 1:300) by co-staining with DAPI for nuclei. After mounting, the sections were scanned by an automatic digital slide scanner and analyzed by the Case Viewer 2.1 software (Pannoramic MIDI, 3D HISTECH, Hungary).

**In vitro osteogenic expression of BMSCs encapsulated in hydrogels.** Rabbit BMSCs were isolated and harvested. Neonatal New Zealand white rabbits (0044063, Chengdu Dossy Experimental Animals Co., LTD., China) were euthanized under sterile conditions. Long bones with femoral heads but without cartilage were collected, and bone marrow was flushed using a syringe with alpha-modified Eagle's medium (α-MEM, 22571038, Gibco) containing 20% fetal bovine serum (FBS, 10100147, Gibco) and antibiotics (penicillin 100 U/mL, streptomycin 100 mg/mL, 10378016, Gibco). A cell strainer (70 mm, 352350, Falcon) was used to remove bone fragments. Filtered bone marrow cells were cultured in 10 cm cell culture dishes at 37 °C in a humidified atmosphere of 5% CO$_2$. Nonadherent cells were removed by changing the medium after 24 h culture. The BMSCs were cultured and passaged until a confluence of 90% was achieved. The BMSCs of the second passage were cultured in a medium with 10% FBS and used in the following experiments. The proliferation of BMSCs encapsulated in various scaffolds was evaluated by CCK-8 (C0037, Beyotime, China) assay with 96-well plates (3595, Costar). After co-culturing for 3, 7, and 14 days, 10% of CCK-8 solution was added into each well for 4 h, and the absorbance value was measured at 450 nm by using a Multiscan Spectrum (Varioskan Flash, Thermo Fisher Scientific, USA). The specimens were taken out on days 3, 7, and 14, washed twice with PBS, and immersed in PBS solution containing 1 μg/mL of fluorescein diacetate (FDA, F1303, Invitrogen) and 1 μg/mL of propidium iodide (PI, P1304MP, Invitrogen), which stain viable cells and dead cells, respectively. The viability and distribution of BMSCs in hydrogels were observed by CLSM (LSM 880, Zeiss) with ZEN software (Carl Zeiss Microscopy GmbH, Version 2.3.69.1000). Before characterizing the morphological change of BMSCs by SEM, specimens with BMSCs were fixed in 2.5% glutaraldehyde solution for 48 h at 4 °C, subsequently dehydrated against a graded series of alcohol and dried by critical point drying for 90 min.

For cytoskeletal staining, samples were firstly permeabilized with 0.1% v/v Triton X-100 for 10 min, and then suffered from rhodamine-phalloidin (5 μg/mL, 1 h, 94072, Sigma-Aldrich) and DAPI (10 μg/mL, 1 min, 28718-90-3, Sigma-Aldrich) staining at 25 °C. For sections staining, samples were washed twice with PBS and soaked in 4% paraformaldehyde (30525-89-4, CHRON CHEMICALS, China) solution for 48 h. Then they were processed for the paraffin section, and the sections (10 μm) were processed for immunohistochemistry staining of BMP-2 (ab284387, Abcam, 1:100), VEGF (ab32152, Abcam, 1:250), and goat anti-rabbit secondary antibody (Abcam, ab155079, 1:200). The mRNA expression level of osteogenic-related genes OCN, OPN, and Runx2 was investigated after co-culturing for 3, 7, and 14 days. Rabbit-Actin was used as the housekeeping gene. Total RNA was extracted from hydrogels (DCLS, HCLS, HAD, and Col I) using an RNeasy Mini Kit (74104, Qiagen) following the manufacturer's instructions. The samples were frozen in liquid nitrogen and ground by glass rod grinding (MP Bio, FastPreP-24). RNA was reverse transcribed into complementary DNA (cDNA) using an iScript cDNA Synthesis Kit (BioRad) according to the reaction protocol. The qRT-PCR reaction was performed using iTaq$^{TM}$ universal SYBR Green Supermix (Biorad) in a CFX96 real-time PCR detection system (BioRad CFX Manager). The primer and probe sequences in real-time RT-PCR reactions were listed in Supplementary Table 1.

**Transcriptome sequencing and data analysis.** BMSCs ($5 × 10^6$ cells/mL) were encapsulated with different samples and co-cultured in a 24-well plate for 14 days. Then, cells were lysed by trizol reagent (R0016, Beyotime Biotechnology), and cell lysates were stored at −80 °C before sequencing. RNA sequencing was performed using an Illumina HiSeq X10 (Illumina, USA). The value of gene expression was transformed as log$_{10}$ [TPM (Transcripts Per Million reads) + 1]. The GO and KEGG pathway enrichment analyses were performed using the free online platform of OE Cloud Platform (cloud.oebiotech.cn).

**BMSCs retention in vitro**. BMSCs suspension was collected and cultured with various scaffolds. After 3, 7, and 14 days of co-culturing, scaffolds were rinsed to purge any unattached cells. Then, live/dead staining was performed to evaluate the viability and distribution of cells by CLSM, and the density of recruited cells on various scaffolds was calculated based on the obtained images. Each group was investigated in triplicate, and three random views were captured for each sample. Furthermore, cytoskeletal staining was carried out by rhodamine-phalloidin and DAPI staining, and SEM was used to further confirm the cell morphology.

**Cell recruitment in vitro and in vivo**. Total rabbit bone marrow was collected from cranial defect area and cultured with various scaffolds. After 48 h culture, scaffolds were rinsed to purge any unattached cells and FITC-phalloidin staining was performed to identify the cells. Furthermore, following the instructions of the manufacturer, CD44 (ab157107, Abcam) immunofluorescence staining was carried out and examined by CLSM. The density of recruited cells on various scaffolds was calculated based on the obtained images. Each group was investigated in triplicate, and three random views were captured for each sample. Recruitment of BMSCs in vitro was measured by a transwell-migration assay (3422, Costar). Briefly, four groups of scaffolds were introduced to a transwell insert's lower chamber, and ~5 × 10³ BMSCs per well were cultured on the transwell insert's upper chamber in 0.1 mL of α-MEM for 12 h. Next, the transwell chamber was fixed in 4% paraformaldehyde and stained with 0.5% crystal violet. The BMSCs not penetrating the filters were swabbed with cotton swabs, and cells that migrated to the lower surface were observed under a light microscope. The results were quantified by dissolving crystal violet in 33% acetic acid and measured at 573 nm on a Microplate reader. Each group was investigated in triplicate, and three random views were captured for each sample.

Effects of various scaffolds on recruitment of ESCs in vivo were assessed. Six adults New Zealand white rabbits (2.5–3.0 kg, male, 2.5–3 months old) were used in this study (0020844, Chengdu Dossy Experimental Animals Co., LTD., China). After anesthetized, scaffolds were transplanted into a 9 mm-diameter defect on each side of the cranial bone. After one week, the SEM, rhodamine-phalloidin, and DAPI staining were used for detecting cell morphology. Besides, samples were embedded in paraffin wax and sectioned at a thickness of 5 μm for H&E, BMP-2 (PA5-88752, Invitrogen, 1:200), Runx2 (Invitrogen, PA5-87299, 1:100), CD31 (MA1-26196, Invitrogen, 1:100), CD90 (Abcam, ab181469, 1:200) and CD44 (Invitrogen, MA5-13890, 1:100) staining. CLSM was used to get image information and quantitative analysis was conducted by dual turntable confocal scanning (CytometeIr CQ1, Yokogawa, Japan)[74,75]. Each group was investigated in triplicate.

**Ectopic osteogenesis of BMSCs-laden hydrogels in nude mice model**. A murine dorsal subcutaneous pocket model was chosen to estimate bone formation on various hydrogels encapsulated BMSCs. After co-culturing with BMSCs in vitro for one week, BMSCs-laden hydrogels were implanted subcutaneously into nude mice (n = 6, D000521, GemPharmatech Co., Ltd, China). Each mouse received dorsal subcutaneous implants (laden 2 × 10⁶ cells) at both sides of the back, and implants were allowed to develop in vivo for 4 weeks. The mouse was euthanized at week 4 post-surgery, and the samples were excised and fixed overnight in 4% paraformaldehyde. The total bone volume formed in hydrogels was visualized and quantified using micro-CT. The degree of bone ingrowth was further determined by the AV/TV (Apatite volume/total volume) ratio. Afterwards, samples were embedded in paraffin wax and sectioned for H&E, VEGF (ab32152, Abcam, 1:150), CD31 (MA1-26196, Invitrogen, 1:100), and Runx2 (Invitrogen, PA5-87299, 1:100) staining. Goat anti-mouse second antibody (Abcam, ab150113, 1:400) was used as the second antibody. Microstructure and mineralized components were characterized by SEM and EDS, while the mechanical property of post-implants was evaluated by DMA.

**Activating skull reconstruction by self-adhesive and flexible scaffold in rabbit model**. The 48 adults New Zealand white rabbits (2.5–3.0 kg, male, 2.5–3 months old, 0044063, Chengdu Dossy Experimental Animals Co., LTD., China) were used in this study. After anesthesia, scaffolds were transplanted into a 9 mm-diameter defect on each side of the cranial bone (n = 12). Animals were grouped as follows: blank (defected only), DCLS, HCLS, and normal (without surgery). Samples were taken out at weeks 4 and 12, then gross appearance and microstructure were photographed by stereomicroscope (SYCOP 3, ZEISS, Germany) and SEM. Push-out test was conducted to evaluate osseointegration performance (Shimadzu, EZ-LX 1 kN) at weeks 4 and 12. Micro-CT analyses were performed as described above. Quantitative analysis of BV/TV, bone mineral density, trabecular thickness, the number of bone trabeculae, and trabecular spacing was determined by micro-CT assistant software. Total soft and bone tissue in the defect area was defined as tissue volume (TV). All samples were radiographed by a digital X-ray unit with 1.8 s exposure at 62 kVP and 250 mA. Osteogenesis-related gene expression was studied at week 12. The gene expression of *VEGF*, *BMP-2*, *OCN*, *Col I*, *OPN*, and *RUNX2* were investigated in this study as described above and the primer sequences were listed in Supplementary Table 2. All data were processed using BioRad CFX Manager software, and the relative gene expression was calculated based on a house-keeping gene, Rabbit-Actin. Furthermore, protein expression of RUNX2 (ab23981, Abcam, 1:1500), BMP-2 (ab284387, Abcam, 1:1000), and VEGF (ab32152, Abcam, 1:1000) were evaluated by WB. Total

proteins were extracted using RIPA buffer (P0013C, Beyotime, China) and the protein concentration of cell extracts was quantitatively analyzed with BCA Protein Assay Kit (Pierce™ Rapid Gold BCA Protein Assay Kit, 23227, Thermo Fisher) according to the kit's instructions. Chemiluminescence was detected and the Alpha Ease FC software processing system was used to analyze the optical density of the target band. The relative content ratio of gray value was used as protein quantitative analysis that each group was investigated in triplicate. For histological analysis, samples were fixed in 4% paraformaldehyde and then immersed in 10% EDTA decalcifying solution (6381-92-6, CHRON CHEMICALS, China), which was changed once a week. After that, samples were paraffin sectioned at a thickness of 5 μm followed by H&E, MT (G1343, Solarbio), and CD31 (Invitrogen, MA5-13188, 1:100) immunofluorescence staining. Immunohistochemical staining was further performed to analyze the osteogenesis-related matrix, and reagents used included BMP-2 (ab284387, Abcam, 1:200), Runx2 (bs-1134R, Bioss, 1:250), OCN (GTX13418, GENETEX, 1:200), VEGF (bs-1313R, Bioss, 1:100), Goat anti-Rabbit IgG (H + L) Cross-Adsorbed Secondary Antibody and HRP (Thermo Fisher Scientific, G-21234, 1:100). Staining sections were observed with a multispectral tissue scanner (Pannoramic midi, 3D HISTECH) and analyzed by the Case Viewer software.

**The skull reconstruction in beagle dog model**. Six adult Beagle dogs (1 year, 6.7–8.5 kg) were used in this study (0020832, Chengdu Dossy Experimental Animals Co., LTD., China). After anesthetizing, the skull was marked and the flaps or muscle flaps were carefully separated by a high-frequency electric knife to avoid cerebrospinal fluid leakage or blade injury to brain tissue. Initial hemostasis was performed with medical gauze, and the wound was cleaned with normal saline. Then, a spreader was used to support both sides of the incision to expose the location of hole. A hand-held dental ball drill (Φ = 2.3 mm, HM1023-round head, Rick ward, Corporation, China) was used to drill holes with a diameter of ~15 mm on both sides of the skull. During the drilling process, saline solution was rinsed every 2 min or so to remove dross and exudate blood while cooling down. Hemostasis was performed with medical gauze and cerebral cotton tablets were covered on the defect, simultaneously physiological saline was added to make it wet. Scaffold was then transplanted into a defect (diameter 15 mm) on one side of cranial bone while the other side was a blank control group. Animals were grouped as follows: blank (defected only) and HCLS (n = 3). To assess new bone formation in defect sites, the specimens were radiographed using digital X-ray unit and Micro-CT as described above. Subsequently, they were processed for paraffin section, and the sections (5 μm) were processed for H&E, MT (G1343, Solarbio), and CD31 (bs-0195R, Bioss, 1:250) immunofluorescence staining.

**Statistical analysis**. At least three times each experiment was repeated independently with similar results and the results could be reproduced according to this method. Statistical analyses were carried out using the SPSS 22.0 software and the results are reported as the mean ± SD (standard deviation). Statistical differences were shown with three significance levels. $*P < 0.05$, $**P < 0.01$, and $***P < 0.001$.

**Reporting summary**. Further information on research design is available in the Nature Research Reporting Summary linked to this article.

## Data availability
The data that support the findings of this study are available within the article and its Supplementary Information files. Raw sequencing data generated in this study have been deposited in the NCBI SRA database under accession number PRJNA773733, and the BioProject's data are now publicly available at "PRJNA773733". Genome Database (OryCun2.0_NCBI) [https://www.ncbi.nlm.nih.gov/genome] and GO Database [http://geneontology.org/] are used in this article. Other data are available from the authors upon reasonable request. The source data underlying Figs. 2–9 and Supplementary Figs. 13, 14, and 30C are provided as a Source Data file. Source data are provided in this paper.

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

## Acknowledgements

This work was supported by the National Key R&D Project of China (Grant No. 2018YFC1106800), National Natural Science Foundation of China (32071352), Sichuan Province Key R&D Project (2019YFS0007), and Sichuan University Innovation Spark Project (2018SCUH0089).

## Author contributions

G.G.L. and Y.X. contributed to the execution and analysis of all experiments. Q.Y.L., M.Y.C, H.S., Y.X.W., and P.L.W. contributed to animal studies and tissue processing. M.Y.C. and X.L. contributed to in vitro cell migration and angiogenesis experiments. X.L., E.L., and X.H.H. contributed to mainly operating large animal surgery. J.L. contributed to the data analysis. Q.J. and J.L. contributed to experimental testing and methods. Y.J.F. and Y.S. conceived the project and contributed to the study design and result analysis. X.D.Z. provided research guidance. G.G.L., Y.X., Y.J.F., and Y.S. prepared the manuscript with inputs from all authors.

## Competing interests

The authors declare no competing interests.
