## [Peer review file · Nature Communications]

REVIEWER COMMENTS

Reviewer #1 (Remarks to the Author):

A Review of NCOMMS-21-12583,

Title: Instantly Fixable and Self-adaptive Membranous Scaffold for Skull Regeneration by Autologous Stem Cell Recruitment and Angiogenesis

This review article mainly focuses on the development of dopamine-mediated hybrid crosslinked membranous scaffold (HCLS) for the usage of facilitated regeneration of skull bone regeneration, especially via stem cell recruitment as well as angiogenesis. In particular, the performance in a dog model is able to support the functionality of developed HCLS platform. However, it is also necessary to improve the quality of the present manuscript, as guided in the following comments.

1. One of the representative properties in HCLS is the dopamine-mediated modification. With the aid of dopamine's adhesion ability, author stated that this material could enhance instant adhesion with surrounding host bone tissues and subsequently facilitate endogenous stem cell recruitment. This localization of stem cell population into the transplanted site might be able to downstream osteogenesis and angiogenesis. Nevertheless, the major engineering concept using dopamine have been reported in numerous previous investigations. Moreover, in terms of main biomaterials used in the current study (i.e., hyaluronic acid, type I collagen, and hydroxyapatite mineral components) do not possess any novelty as compared with previously reported craniofacial implantable materials for skull bone regeneration.
2. Another major issue (also found in many studies about cell free material-based implants) is a lack of systemic investigations of underlying mechanisms to reveal how the bulk properties of implantable materials precisely regulate and control molecular levels of cellular responses. Although the bulk properties of instantly fixable and self-adaptive characteristics in HCLS could facilitate a proper interaction with host tissues and eventual upregulation of stem cell infiltration, a precise and sequential confirmation to reveal how these material-mediated intrinsic mechanical properties control a series of cellular/molecular phenomena.
3. In terms of visual performance in submitted video file, the morphology of HCLS is not membranous due to its height.

Reviewer #2 (Remarks to the Author):

This research address an important area in the use of biomaterials for the treatment of craniofacial bone defects. While the project is in a particularly crowded space, there are some particularly interesting features of the biomaterial used- an adhesive and flexible hybridized cross-linked scaffold (HCLS) fabricated by dopamine (DA)-modified hyaluronic (HA) crosslinked with type I collagen (Col I) and micron sized hydroxyapatite particles.

The objectives are reasonably ambitious but do not greatly move beyond the state of the art although the authors are to commended on just how comprehensive the study is with an extensive range of in vitro and in vivo experiments carried out to both characterise the materials developed and demonstrate their efficacy for bone repair. Generally the results show significant promise and demonstrate the potential of the materials to facilitate osteogenesis and angiogenesis and the materials show efficacy in healing bone defects- particularly in the rabbit defect. Saying that, there are a number of issues outstanding before a recommendation for publication should be considered.

These include:

- The rationale for the material design and its novelty over the plethora of under biomaterials for bone repair needs to be emphasised further.
- Considering the depth of biological characterisation carried out, it's somewhat surprising that the mechanical characterisation is somewhat lacking in comparison. More information is required on mechanical properties in terms of moduli and flexural properties - particularly as flexible properties are proposed as one of the selling points of the material. Some crude images are presented in the supplementary information but more information is needed.
- While the angiogenic and osteogenic properties are very well characterised in vitro and in vivo, an area that is not sufficiently analysed is the immune response. To fully characterise the material, expanded information is required- including assessment of the response of the materials to macrophages in vitro and more thorough assessment of the in vivo immune response. For example, it would be useful to know whether BMSC & ESC recruitment was connected with macrophage activity. Similarly, the authors describe 'No obvious inflammatory response was observed in all groups' in the rabbit model and that 'the gross view showed there were no adverse reactions such as redness, suppuration or necrosis of tissue around implant area (red circles), indicating good compatibility...' in the dog model. However, a more thorough assessment is required beyond gross assessment including assessment of whether a predominant M2 macrophage response was seen as healing progressed.

In the dog model, some promise is seen but the results are not as compelling as in the rabbit models. The authors note that 'the structural design of implant would be further optimized to match the mechanical environment of defect sites.' This goes back to my earlier question on mechanical properties

where further information is needed to compare the mechanical properties of the HCLS implant to the bony environment. Furthermore, more information on rabbit, dog and indeed human craniofacial bone properties might enhance this discussion.

Also, for consistency and comparison with the rabbit study, the authors should present bone cover area (97% in rabbit cranial defects) but presumably, judging the images presented, much lower in dogs?

Finally, the language in the paper needs careful editing as the language is not sufficiently fluent in many areas. In addition, too many terms such as 'amazing' and 'remarkably' are used when describing the results. This should be toned down a little.

Reviewer #3 (Remarks to the Author):

The authors developed a hybrid cross-linked membranous scaffold (HCLS) using dopamine-modified hyaluronic acid (HAD), type I collagen (Col I), and micron hydroxyapatite (μ HAp). The study was designed comprehensively to assess the potential efficiencies of HCLS for skull regeneration. The results demonstrated that the HCLS could promote the proliferation and osteogenic differentiation of bone mesenchymal stem cells (BMSCs) and recruit endogenous stem cells (ESCs) to initiate osteogenesis and angiogenesis in both rabbit and beagle dog cranial defect models. The key point of the study is the instantly fixable and self-adaptive properties of HCLS, yet the using of HAD is a simple and universal surface modification method inspired by adhesion ability of mussels, which has been widely used in recent years. The novelty of the study needs to be improved. Some other points and suggestions are addressed below.

Major:

(1) In lines 119-120, the authors used DAPI staining to confirm the uniform distribution of μ HAp in HCLS. However, DAPI is a fluorescent dye that binds to DNA. How can μ HAp be identified by DAPI staining?

(2) In lines 140-145, the references 30 and 31 might be inappropriate to expound the impact of μ HAp on the increase of mechanical strength of HCLS hydrogel, as μ HAp and graphene oxide possess different structures and functional groups. Besides, the scanning electron microscope (SEM) results of Fig. 1C were insufficient to draw a conclusion that μ HAp were cross-linked in the HCLS.

(3) In lines 212-218, in the experiment of in vitro cell recruitment, the HCLS was incubated with the whole rabbit cranial bone marrow cell suspension, and its ability to recruit BMSCs was affirmed by CD44+ cells. However, the bone marrow contains many different types of cells, including hematopoietic stem cells (HSCs) and BMSCs, and only about 0.001-0.01% of the cells in the bone marrow are BMSCs. Besides, CD44 is first described for HSCs. Thus, it was imprecise to conclude that the recruited cells were BMSCs only by CD44 staining. The addition of other markers, such as CD90, can be used to identify BMSCs.

(4) In Fig. 7B2-B3, the BV/TV value and new bone volume of the Blank group at 12 weeks were both lower than those at 4 weeks, please explain the reasons for these results.

Minor:

(1) In line 102, the description of the absorption peak at 280 nm was inconsistent with that in Supplementary Fig. 2C.

(2) Please check the statistic difference between the HCLS and HAD groups in Fig. 1F.

(3) In line 171, the storage modulus of DCLS (59 ± 5 kPa) on 30th day was inconsistent with that in Fig. 3B2.

(4) In lines 174-185, the serial number of the Supplementary Figure was misquoted, 'Supplementary Fig. 8' should be 'Supplementary Fig. 7'. Likewise, 'Supplementary Fig. 9' should be 'Supplementary Fig. 8' (line 190 and line 197).

(5) Please confirm the number of defect in each beagle dogs. Only one defect were drew in the figure Scheme 1 (line 763), while described in the method as 'scaffolds were transplanted into a defect on each side of the cranial bone' (line 623).

Dear Editor and Referees,

We highly appreciate the valuable comments and suggestions of reviewers, which greatly helped us to improve the quality of our manuscript. According to these comments and suggestions, we revised the manuscript carefully.

We hereby, on behalf of the co-authors, submitted the revised manuscript entitled “Instantly fixable and self-adaptive scaffold for skull regeneration by autologous stem cell recruitment and angiogenesis” to “**Nature Communications**” for your consideration, which included the Manuscript (8 Figures and 1 Scheme) and Supporting Information (26 Figures, 2 Videos and 2 Tables). The source data underlying Figure 1-8 and Supplementary Figure 8, 9 and 25C were provided as a Source Data file.

The point-to-point response to the reviewers’ comments was listed as follows:

Response to Reviewer #1 (Remarks to the Author):

A Review of NCOMMS-21-12583,

Title: Instantly Fixable and Self-adaptive Membranous Scaffold for Skull Regeneration by Autologous Stem Cell Recruitment and Angiogenesis

This review article mainly focuses on the development of dopamine-mediated hybrid crosslinked membranous scaffold (HCLS) for the usage of facilitated regeneration of skull bone regeneration, especially via stem cell recruitment as well as angiogenesis. In particular, the performance in a dog model is able to support the functionality of developed HCLS platform. However, it is also necessary to improve the quality of the present manuscript, as guided in the following comments.

Q1. One of the representative properties in HCLS is the dopamine-mediated modification. With the aid of dopamine’s adhesion ability, author stated that this material could enhance instant adhesion with surrounding host bone tissues and subsequently facilitate endogenous stem cell recruitment. This localization of stem cell population into the transplanted site might be able to downstreaming osteogenesis and angiogenesis. Nevertheless, the major engineering concept using dopamine have been reported in numerous previous investigations. Moreover, in terms of main biomaterials used in the current study (i.e., hyaluronic acid, type I collagen, and hydroxyapatite mineral components) do not possess any novelty as compared with previously reported craniofacial implantable materials for skull bone regeneration.

Reply: As shown in Fig. 1, dopamine-modified hyaluronic acid (HAD) acted as a “bridge” to chelate Ca^{2+} in the surface of micron hydroxyapatite (μHAp) and bind type I collagen (Col I) by Michael addition reaction. This strategy effectively integrated organic and inorganic phases with strong chemical connection in molecular level, which mimicked the construction of natural bony ECM. As the results, the obtained hybrid cross-linked scaffold (HCLS) could maintain structural integrity through flexible deformation to respond external stress in the defect, and present well mechanical match and interface integration with bony host environment by tissue adhesion. Furthermore, the proper immune microenvironment provided by HCLS could regulated macrophages polarization to M2 phenotype. After implantation in both rabbit and beagle dog cranial defect models, HCLS could rapidly initiate angiogenesis and osteogenesis by *in-situ* recruitment of ESCs into defect sites, and subsequently accelerate osteo-differentiation to regenerate skull. The corresponding descriptions have been added to the revised manuscript and were marked as red.

Q2. Another major issue (also found in many studies about cell free material-based implants) is a lack of systemic investigations of underlying mechanisms to reveal how the bulk properties of implantable materials precisely regulate and control molecular levels of cellular responses. Although the bulk properties of instantly fixable and self-adaptive characteristics in HCLS could facilitate a proper interaction with host tissues and eventual upregulation of stem cell infiltration, a precise and sequential confirmation to reveal how these material-mediated intrinsic mechanical properties control a series of cellular/molecular phenomena.

Reply: In revised manuscript, transcriptomic analysis of BMSCs cultured *in vitro* on HCLS, DCLS and Col I for 14 days was applied to reveal how the bulk properties of implantable materials precisely regulate and control molecular levels of cellular responses. The Pearson correlation and principal component analysis were used to assess the specimen’s stability. The correlation coefficients of samples in each group were within acceptable ranges ($R^2 > 0.94$, $N= 3$), indicating well biological repetition. While the value of samples among different groups also illustrated obvious gene expression differences in HCLS vs Col I and DCLS vs Col I ($R^2 < 0.85$, $N= 3$), as well as potential similarity in HCLS vs DCLS ($R^2 > 0.94$, $N=3$) (Fig. 5A1, A2). There were many gene expression differences (1502 genes in HCLS vs DCLS, 2240 genes in HCLS vs Col I, and 2378 genes in DCLS vs Col I,), and they shared 12427 genes (Fig. 5A3). The number of differentially expressed genes between HCLS vs Col I and DCLS vs Col I was not significant. Thus, HCLS vs DCLS and DCLS vs Col I groups were used as pairwise comparison to analyze the differences

caused by introduction of HAD and μ HAp. As shown in Fig. 5B1-B4 and C1-C4, the volcano plots and heatmaps showed 366 up-regulated and 133 down-regulated genes (DCLS vs Col I), 58 up-regulated and 29 down-regulated genes (HCLS vs DCLS) involved in the top enriched up-GO (Gene Ontology) terms. Besides, 207 up-regulated and 70 down-regulated genes (DCLS vs Col I), 19 up-regulated and 5 down-regulated genes (HCLS vs DCLS) were found in the top enriched up-KEGG (Kyoto Encyclopedia of Genes and Genomes) pathways, respectively.

The differentially expressed genes were collected and typically divided into four typical categories, including cellular activity, osteogenesis, angiogenesis, and ESCs recruitment. Significant differences of each category could also be found from volcano plots and heatmaps in DCLS vs Col I (Supplementary Fig. 22A1-4, B1-4) and HCLS vs DCLS groups (Supplementary Fig. 23A1-4, B1-4), while their differentially expressed genes for each category were also respectively collected to perform GO database analysis (Supplementary Fig. 22C1-4 and Fig. 23C1-4). Generally, the upregulated genes in DCLS vs Col I (Fig. 5D1) were enriched orderly in ECM, collagen binding, chemotaxis, signaling receptor activity, cell migration, cell adhesion, angiogenesis and ossification. Compared with DCLS, the HCLS (Fig. 5D2) further enhanced the related genes expression in ECM structure and organization, integrin binding, epidermal cell differentiation, ossification, peptide binding, vasculogenesis and labyrinthine layer blood vessel development.

The relevant top enriched up-KEGG pathways speculated that HAD in DCLS (Fig. 5D3) could not only increase cell migration and adhesion (cytokine-cytokine receptor interaction, chemokine signaling pathways, cell adhesion molecules, focal adhesion), but also improve proinflammatory expression and M2 macrophage polarization (NF- κ B and peroxisome proliferator-activated receptor (PPAR) signaling pathways).⁴⁷ Besides, DCLS significantly promoted stem cells development and matrix formation (signaling pathways regulating pluripotency of stem cell, ECM-receptor interaction, MAPK signaling pathway), and subsequently enhance angiogenesis (Hematopoietic cell lineage) and osteogenesis (PI3K-Akt signaling pathway, HIF-1 signaling pathways and TGF- β signaling pathways) in relative to Col I. Furthermore, μ HAp particles in HCLS (Fig. 5D4) would accelerated osteogenesis (highest enrichment in PI3K-Akt signaling pathway)⁴⁸ by promoting cell adhesion (Focal adhesion, Mucin type O-glycan biosynthesis, Rap 1 signaling pathway) and activity (Cell cycle, ECM-receptor interaction)⁴⁹, as well as potential inflammatory regulation (Phospholipase D signaling pathway, enhancing the transcriptional

activity of NF- κ B, and promoting the transcription and expression of cyclin D1 and vascular endogenesis factor).⁵⁰ Specific genes heat maps and their corresponding protein-protein interaction networks clarified the process, confirmed that the introduction of HAD and μ HAp might played a positive role in recruiting ESCs by activating expression and interaction of osteogenic/angiogenic related proteins (Fig. 5E1-E4). The transcriptomic data analysis was consistent with previous experimental results in Fig. 2-Fig. 4, which indicated that by integrating HAD and μ HAp, HCLS might accelerate integrins/peptides/factors binding to recruit ESCs, and then perfect ECM structure and organization, promote vasculogenesis of advanced structures and more intensely and timely osteogenesis. The corresponding descriptions and results have been added to the revised manuscript and were marked as red.

Q3. In terms of visual performance in submitted video file, the morphology of HCLS is not membranous due to its height.

Reply: HCLS was a flexible adherent porous scaffold with thin layer morphology. In submitted video file, in order to match the height of skull defect, we controlled the thickness of HCLS at about 2 mm. According to preparation method, we could easily modulate the thickness based on specific application requirements. In order to avoid misunderstanding, we deleted “membranous” in the revised manuscript.

Response to Reviewer #2 (Remarks to the Author):

This research address an important area in the use of biomaterials for the treatment of craniofacial bone defects. While the project is in a particularly crowded space, there are some particularly interesting features of the biomaterial used- an adhesive and flexible hybridized cross-linked scaffold (HCLS) fabricated by dopamine (DA)-modified hyaluronic (HA) crosslinked with type I collagen (Col I) and micron sized hydroxyapatite particles.

The objectives are reasonably ambitious but do not greatly move beyond the state of the art although the authors are to commended on just how comprehensive the study is with an extensive range of in vitro and in vivo experiments carried out to both characterise the materials developed and demonstrate their efficacy for bone repair. Generally the results show significant promise and demonstrate the potential of the materials to facilitate osteogenesis and angiogenesis and the materials show efficacy in healing bone defects- particularly in the rabbit defect. Saying that, there are a number of issues outstanding before a

recommendation for publication should be considered.

These include:

Q1. The rationale for the material design and its novelty over the plethora of under biomaterials for bone repair needs to be emphasized further.

Reply: As shown in Fig. 1, dopamine-modified hyaluronic acid (HAD) acted as a “bridge” to chelate Ca^{2+} in the surface of micron hydroxyapatite (μHAp) and bind type I collagen (Col I) by Michael addition reaction. This strategy effectively integrated organic and inorganic phases with strong chemical connection in molecular level, which mimicked the construction of natural bony ECM. As the results, the obtained hybrid cross-linked scaffold (HCLS) could maintain structural integrity through flexible deformation to respond external stress in the defect, and present well mechanical match and interface integration with bony host environment by tissue adhesion. Furthermore, the proper immune microenvironment provided by HCLS could regulate macrophages polarization to M2 phenotype. After implantation in both rabbit and beagle dog cranial defect models, HCLS could rapidly initiate angiogenesis and osteogenesis by *in-situ* recruitment of ESCs into defect sites, and subsequently accelerate osteo-differentiation to regenerate skull. The corresponding descriptions have been added to the revised manuscript and were marked as red.

Q2. Considering the depth of biological characterization carried out, it's somewhat surprising that the mechanical characterization is somewhat lacking in comparison. More information is required on mechanical properties in terms of moduli and flexural properties - particularly as flexible properties are proposed as one of the selling points of the material. Some crude images are presented in the supplementary information but more information is needed.

Reply: Mechanical and wet adhesion properties of scaffold were further investigated following the suggestion. According to tensile tests, HCLS achieved the highest tensile force of 34.8 KPa at breakdown point, and it could be folded like origami and unfolded into flat sheet (Figs. 1H1, 3 and Supplementary Video 1). The mechanical properties of scaffolds with blood infiltration (simulating *in vivo* application environment) were further evaluated. Although the tensile stress of scaffolds decreased dramatically with blood infiltration from 4.2 to 0.3 N for DCLS and 5.6 to 0.7 N for HCLS, respectively, the deformation range obviously elongated (Fig. 1I, Supplementary Fig. 5A). Dynamic mechanics (DMA)

tests results indicated that the storage modulus (G') of DCLS declined markedly after blood infiltration (55 to 35 KPa), but no significant differences of G' were observed in HCLS (90 to 85 KPa), with hemoinfiltration (Fig. 1J1-3, Supplementary Fig. 5B). The HCLS could be instantly fixed to cranial defect site in wet environment (Fig. 1H2 and Supplementary Video 2), which mainly attributed to high affinity of catechol groups to diverse nucleophiles (eg, amines, thiols, imidazoles and chelation of metal particles), providing potential mechanism for tissue adhesion by anchoring to proteins of skull surface.³² As demonstrated in Fig. 1K and Supplementary Fig. 6, the adhesion strength of HCLS achieved the highest against glass, titanium (Ti), skull, pigskin and polyethylene (PE).

The interfacial bonding between scaffolds and host bone was also evaluated in rabbit skull defect model ($\Phi = 9$ mm) after 7 and 14 days implantation. The interface was well integrated at day 7, and became more tightly bound at day 14 by gross observation and SEM (Fig. 1L1 and L2). The interfacial binding force increased over time, and the highest push-out force was achieved in HCLS at day 14 (32 ± 3 N) (Fig. 1L3 and L4). Subsequently, the mechanical properties of the implants were examined by DMA. No obvious size change occurred after 14 days implantation, but the G' of implants rose up to 60 ± 3 KPa in HCLS on day 14 (Fig. 1M and Supplementary Fig. 7A and B), demonstrating satisfactory structural stability and interface integration.

Q3. While the angiogenic and osteogenic properties are very well characterised in vitro and in vivo, an area that is not sufficiently analysed is the immune response. To fully characterise the material, expanded information is required- including assesment of the response of the materials to macrophages in vitro and more thorough assessment of the in vivo immune response. For example, it would be useful to know whether BMSC & ESC recruitment was connected with macrophage activity. Similarly, the authors describe 'No obvious inflammatory response was observed in all groups' in the rabbit model and that 'the gross view showed there were no adverse reactions such as redness, suppuration or necrosis of tissue around implant area (red circles), indicating good compatibility...' in the dog model. However, a more thorough assessment is required beyond gross assessment including assessment of whether a predominant M2 macrophage response was seen as healing progressed.

Reply: Ideal biomaterials for tissue regeneration should induce a controlled inflammatory response, because the inflammation formed at the early stage of wound repair was beneficial for the recruitment of inflammatory cells, biochemical factors, and bone progenitor cells. Implantable biomaterials with

favorable immune responses could guide the success of osteogenesis and angiogenesis.

In the revised manuscript, the immune response of various scaffolds was firstly evaluated *in vitro*. Compared with the smooth boundaries of RAW 264.7 macrophages on DCLS, abundant filopodia with long stretching distances were found on HCLS by SEM. The quantitative results showed a higher spreading area and cell aspect ratio on HCLS (Supplementary Fig. 8A, C1, C2), implying the promoted cell spreading and distinctive cell morphologies by μ HAp. Immunofluorescent staining was applied to visually analyze immunological polarity induced by HCLS and DCLS (Supplementary Fig. 8B). The semi-quantitative immunofluorescence intensity by Image J showed that HCLS exhibited higher CD206⁺ expression (highly specific M2 type marker) and lower CD197⁺ expression (highly specific M1 type marker) than that of DCLS (Supplementary Fig. 8C3, C4).

H&E analysis showed that infiltration of inflammatory cells mainly concentrated on the periphery of both DCLS and HCLS at day 7 after intramuscular implantation in mouse, and fibrous tissue were found penetrating into the interior zone of implants at day 14 (Fig. 2A and Supplementary Fig. 9A). A large number of cells were observed both in HCLS and DCLS, but HCLS presented a higher spreading area and cell aspect ratio than DCLS (Fig. 2B and Supplementary Fig. 9B1 and B2). Flow cytometry data indicated that macrophages levels in HCLS was less than that in DCLS, and decreased over time (Fig. 2C1, C2). HCLS recruited higher numbers of M2 macrophages (CD197⁻CD206⁺ cells) and less M1 macrophages (CD197⁺CD206⁻ cells) (Fig. 2C3, C4). The immune-related cytokines *in vivo* were examined by using ELISA to further investigate the inflammatory responses (Fig. 2D1-5). Compared with DCLS, HCLS down-regulated proinflammatory cytokines (TNF- α and IL-1 β), and up-regulated anti-inflammatory cytokines (IL-4, IL-10 and IL-1 α), and this effect became more obvious over time. Immunofluorescence staining further confirmed that HCLS was more conducive to the macrophages M1-to-M2 shift in relative to DCLS (Fig. 2E1-3 and Supplementary Fig. 12). Macrophages were spatially distributed around implants at day 7, whereas more macrophages penetrated into implants at day 14. (Supplementary Fig. 11).

Macrophages could mediate bone and vascular regeneration by secreting osteoinductive and angiogenic factors,^{33,34} such as BMP-2 and VEGF as potent inducers of osteogenesis and angiogenesis.^{35,36} Immunofluorescent staining was applied to explore whether HCLS contributed to osteogenesis and angiogenesis (Fig. 2E1, E4, E5 and Supplementary Fig. 12). It was found that most endogenous VEGF and BMP-2 co-labeled with F4/80⁺ macrophages appeared in implants. The expression

level increased over time both in DCLS and HCLS, but fluorescence intensity in HCLS was obviously higher than that in DCLS. A similar phenomenon occurred in rabbit skull defect model (Fig. 2F1-5 and Supplementary Fig. 13 and 14). Notably, HCLS presented stronger proinflammatory properties (CD197⁺CD206⁻) after 7 days implantation, but the inflammation significantly reduced at day 14. It was reported that the inflammation at the early stage of wound repair was beneficial for the recruitment of inflammatory cells, biochemical factors and bone progenitor cells.³⁷ Hence, higher BMP-2 and VEGF secretion induced by HCLS could be attributed to polarize M2 phenotype macrophages.^{38,39} The corresponding descriptions and results have been added to the revised manuscript and were marked in red.

Q4. In the dog model, some promise is seen but the results are not as compelling as in the rabbit models. The authors note that 'the structural design of implant would be further optimized to match the mechanical environment of defect sites. This goes back to my earlier question on mechanical properties where further information is needed to compare the mechanical properties of the HCLS implant to the bony environment. Furthermore, more information on rabbit, dog and indeed human craniofacial bone properties might enhance this discussion.

Reply: The skull was a non-load-bearing bone tissue, although the modulus of HCLS was relatively low (G'~30 KPa), it could form acceptable mechanical match and good interface integration with surrounding host bony environment in defect site by instantly fixable and self-adaptive properties. With the extension of implantation time, HCLS could rapidly induce bone regeneration and enhance its mechanical strength through rapidly initiating angiogenesis and osteogenesis. After 12 weeks implantation, the HCLS could realize extensive bone regeneration with bone cover area at 97% for rabbit model ($\Phi = 9$ mm) and $71.7 \pm 6.9\%$ for beagle dog model ($\Phi = 15$ mm) at cranial defects sites.

According to literature reports,⁶⁸⁻⁷⁰ the thickness of rabbits, beagle dogs and human skull was about 2 mm, 3 mm and 10~15 mm, and the corresponding elastic moduli was 1~2 MPa, 7.5 GPa and 8~15 GPa, respectively. In addition, rich muscles in beagle skull enhanced the mechanical environment of defect area. Herein, the reasons for this difference could be conjectured as follows: The highly developed muscular tissue on skull of beagle dogs exerted great pressure on the soft implant materials, leading to the structural instability and excessive deformation; in the absence of exogenous fixation, the adhesive force of HCLS might not be enough to support long-term fixation at the defect sites of highly moving beagle dogs. In future research, the structural design of implant would be further optimized to match the mechanical

environment of defect sites. The corresponding descriptions and results have been added to the revised manuscript and were marked as red.

68. Alaqeel, S. M., Hinton, R. J. and Opperman, L. A. Cellular response to force application at craniofacial sutures. *Orthod. Craniofac. Res.* **9**, 111-122 (2010).

69. Radhakrishnan, P., and Mao. J. J. Nanomechanical properties of facial sutures and sutural mineralization front. *J. Dent. Res.* **83**, 470 (2004).

70. Kawahara, H. et al. Osseointegration under immediate loading: biomechanical stress-strain and bone formation-resorption. *Implant Dent.* **12**, 61-68 (2003).

Q5. Also, for consistency and comparison with the rabbit study, the authors should present bone cover area (97% in in rabbit cranial defects) but presumably, judging the the images presented, much lower in dogs?

Reply: Thanks for your valuable comments. As mentioned in the Abstract, after 12 weeks implantation, the HCLS could realize bone regeneration with bone cover area of 72% for beagle dog model ($\Phi = 15$ mm) at cranial defects sites. The corresponding descriptions and results have been added to the revised manuscript and were marked as red.

Q6. Finally, the language in the paper needs careful editing as the language is not sufficiently fluent in many areas. In additon, too many terms such as 'amazing' and 'remarkably' are used when describing the results. This should be toned down a little.

Reply: Thanks for your valuable comments. According to your suggestion, we have carefully revised the article.

Response to Reviewer #3 (Remarks to the Author):

The authors developed a hybrid cross-linked scaffold (HCLS) using dopamine-modified hyaluronic acid (HAD), type I collagen (Col I), and micron hydroxyapatite (μ HAp). The study was designed comprehensively to assess the potential efficiencies of HCLS for skull regeneration. The results demonstrated that the HCLS could promote the proliferation and osteogenic differentiation of bone mesenchymal stem cells (BMSCs) and recruit endogenous stem cells (ESCs) to initiate osteogenesis and

angiogenesis in both rabbit and beagle dog cranial defect models. The key point of the study is the instantly fixable and self-adaptive properties of HCLS, yet the using of HAD is a simple and universal surface modification method inspired by adhesion ability of mussels, which has been widely used in recent years.

Q1. The novelty of the study needs to be improved.

Reply: As shown in Fig. 1, dopamine-modified hyaluronic acid (HAD) acted as a “bridge” to chelate Ca^{2+} in the surface of micron hydroxyapatite (μHAp) and bind type I collagen (Col I) by Michael addition reaction. This strategy effectively integrated organic and inorganic phases with strong chemical connection in molecular level, which mimicked the construction of natural bony ECM. As the results, the obtained hybrid cross-linked scaffold (HCLS) could maintain structural integrity through flexible deformation to respond external stress in the defect, and present well mechanical match and interface integration with bony host environment by tissue adhesion. Furthermore, the proper immune microenvironment provided by HCLS could regulated macrophages polarization to M2 phenotype. After implantation in both rabbit and beagle dog cranial defect models, HCLS could rapidly initiate angiogenesis and osteogenesis by *in-situ* recruitment of ESCs into defect sites, and subsequently accelerate osteo-differentiation to regenerate skull. The corresponding descriptions have been added to the revised manuscript and were marked as red.

Some other points and suggestions are addressed below. Major:

Q2. In lines 119-120, the authors used DAPI staining to confirm the uniform distribution of μHAp in HCLS. However, DAPI is a fluorescent dye that binds to DNA. How can μHAp be identified by DAPI staining?

Reply: DAPI (4',6-diamidino-2-phenylindole) is generally used for staining DNA since it could bind to DNA. But it could also bind to the surface of μHAp through physical adsorption.²⁹ Therefore, we employed CLSM to reconstruct μHAp distribution in HCLS. In addition, micro-CT was used to characterize the sample, and high-resolution 3D reconstruction images were used to characterize the distribution of μHAp , as shown in supplementary Fig. 16A. The corresponding descriptions have been added to the revised manuscript and were marked as red.

29. Shao, C. Y., et al. Citrate Improves Collagen Mineralization via Interface Wetting: A Physicochemical Understanding of Biomineralization Control. *Adv. Mater.* **30**, 1704876 (2018).

Q3. In lines 140-145, the references 30 and 31 might be inappropriate to expound the impact of μ HAp on the increase of mechanical strength of HCLS hydrogel, as μ HAp and graphene oxide possess different structures and functional groups. Besides, the scanning electron microscope (SEM) results of Fig. 1C were insufficient to draw a conclusion that μ HAp were cross-linked in the HCLS.

Reply: This result “the μ HAp were cross-linked in the HCLS” was mainly verified by the following two aspects. Firstly, DSC test results (Fig. 1B) revealed that HCLS achieved denaturation temperature at 113.5°C in relative to 84.6°C of Col I, 88.2°C of HAD, 111.7°C of DCLS and 81.6°C of HA, as well as 100.6°C of HA-Col I- μ HAp and 94.7°C of HA-Col I without introduction of dopamine, suggesting that the successfully hybrid cross-linked strategy enhanced heat stability of HCLS. Secondly, in FTIR of HAD, the peak at 1731 cm^{-1} represented phenolic hydroxyl. In FTIR of HCLS, the 1731 cm^{-1} peak disappeared, indicating the reaction of DOPA with inorganic μ HAp through phenol-quinone transition and Michael addition between quinone and amino groups of Col I. The content of μ HAp in HCLS was 19% w/w from TG analysis (Fig. 1C). The updated references 30 and 31 were applicable to explain adhesion of DOPA on different surfaces. The corresponding descriptions and results have been added to the revised manuscript and were marked as red.

30. Holten-Andersen, N. et al. Metals and the integrity of a biological coating: The cuticle of mussel byssus. *Langmuir*. **25**, 3323-3326 (2009).

31. Ryu, J. et al. Mussel-inspired polydopamine coating as a universal route to hydroxyapatite crystallization. *Adv. Funct. Mater.* **20**, 2132-2139 (2010).

Q4. In lines 212-218, in the experiment of *in vitro* cell recruitment, the HCLS was incubated with the whole rabbit cranial bone marrow cell suspension, and its ability to recruit BMSCs was affirmed by CD44⁺ cells. However, the bone marrow contains many different types of cells, including hematopoietic stem cells (HSCs) and BMSCs, and only about 0.001-0.01% of the cells in the bone marrow are BMSCs. Besides, CD44 is first described for HSCs. Thus, it was imprecise to conclude that the recruited cells were BMSCs only by CD44 staining. The addition of other markers, such as CD90, can be used to identify BMSCs.

Reply: According to the comments, it was imprecise to conclude that the recruited cells were BMSCs only by CD44 staining in the experiment of *in vitro* cell recruitment. In order to confirm that the recruited cells were more likely to be BMSCs, additional immunofluorescence staining (CD90⁺) was further performed

on the samples after one week implantation in rabbit cranial defect site. As shown in revised Supplementary Figure 21, more obvious positive expression of CD90 marker could be found in HCLS in comparison to DCLS. Besides, similar to CD44 staining, the number of cells with CD90 positive expression on the edges of HCLS was more than that in the middle, suggesting that BMSCs migration was also mainly from edge to middle after implantation. The corresponding descriptions and results have been added to the revised manuscript and were marked as red.

Q5. In Fig. 7B2-B3, the BV/TV value and new bone volume of the Blank group at 12 weeks were both lower than those at 4 weeks, please explain the reasons for these results.

Reply: As important processes of bone reconstruction, bone resorption and bone growth coexist at dynamic balance state, which are closely related to the mechanical environment of implant site.⁵⁷ The bone resorption is closely related to osteoclasts. Active osteoclasts at relatively low stress level will secrete acids and enzymes to decompose and absorb the mineralized bone matrix, resulting in decreased bone mass in this area.⁵⁸ In skull reconstruction experiment of beagle dog, compared with HCLS group, the blank group had a lower stress level in defect site due to lack of mechanical support, leading to a gradual decrease in BV/TV value and new bone volume on week 12. The corresponding descriptions and results have been added to the revised manuscript and were marked as red.

57. Shintaro, N &Teruko, T.Y. Molecular events caused by mechanical stress in bone. *Matrix Biology*. **19**. 91-96 (2000).

58. Yin, X., Zhou, C., Li, J. et al. Autophagy in bone homeostasis and the onset of osteoporosis. *Bone Res.* **7**, 28 (2019).

Q6. Minor:

(1) In line 102, the description of the absorption peak at 280 nm was inconsistent with that in Supplementary Fig. 2C.

Reply: Thanks for your valuable comments. According to supplementary Fig. 2C, the maximum absorption peak of HCLS was corrected to 277 nm.

(2) Please check the statistic difference between the HCLS and HAD groups in Fig. 1F.

Reply: Thanks for your valuable comments. As shown in revised Fig. 1E3, there was no statistical

difference between HCLS and HAD groups, while the statistical difference between the HCLS and Col I groups was * $P < 0.05$.

(3) *In line 171, the storage modulus of DCLS (59 ± 5 kPa) on 30th day was inconsistent with that in Fig. 3B2.*

Reply: Thanks for your valuable comments. As shown in revised Fig. 3H3, the compressive storage modulus of DCLS on day 30 was corrected to (65 ± 4 KPa).

(4) *In lines 174-185, the serial number of the Supplementary Figure was misquoted, 'Supplementary Fig. 8' should be 'Supplementary Fig. 7'. Likewise, 'Supplementary Fig. 9' should be 'Supplementary Fig. 8' (line 190 and line 197).*

Reply: Thanks for your valuable comments. We must apologize for making such mistakes, which have been modified according to specific content in revised manuscript. The corresponding descriptions have been added to the revised manuscript and were marked as red.

(5) *Please confirm the number of defects in each beagle dogs. Only one defect was drawn in the figure Scheme 1 (line 763), while described in the method as 'scaffolds were transplanted into a defect on each side of the cranial bone' (line 623).*

Reply: Thanks for your valuable comments. As shown in revised supplementary Fig. 26A, according to the surgical procedure, after anesthetizing Beagle dogs, scaffold (HCLS) was transplanted into defect site (diameter 15 mm) on one side of cranial bone, while the other side was as blank control group. In Scheme 1, only one defect was drawn for schematic implantation process.

I expect your comments on this revised manuscript at your earliest convenience.

Thanks and all the best.

Yours sincerely,

Yong Sun, Ph. D and Yujiang Fan, Ph. D

Signature:

National Engineering Research Center for Biomaterials

Sichuan University

Chengdu, China

REVIEWER COMMENTS

Reviewer #1 (Remarks to the Author):

A Review of NCOMMS-21-12583A,

Title: Instantly Fixable and Self-adaptive Scaffold for Skull Regeneration by Autologous Stem Cell Recruitment and Angiogenesis

Even after the extensive evaluation, the revised manuscript cannot be recommended for the publication. All reviewers questioned the novelty of the present study, the principle of material design, and related scientific rationales, specifically as compared with a number of previous studies using (1) dopamine modification, (2) conventional polymeric & inorganic composites for (3) skull bone regeneration. In spite of the additional transcriptomic & immunological analysis, the major drawback of the current study is still insufficient verification of novelty in materials design along with discrete significances. Only Figure 1A describes a brief chemistry for incorporation of hyaluronic acid and hydroxyapatite, with the aid of dopamine and calcium chelating. However, it is still ambiguous to confirm this approach and resulted material design is superior for the publication in Nature Communications.

Reviewer #2 (Remarks to the Author):

Generally the authors have addressed all my scientific comments and the manuscript is improved by the inclusion of additional data.

However, the language in paper still needs careful editing as is not sufficiently fluent in English - and the new text added is particularly poor. If this is addressed, I'm happy to recommend publication.

Reviewer #3 (Remarks to the Author):

The authors added several experiments to support their conclusions, and the manuscript was improved, but there are details that remain poorly addressed.

1. The authors stated that DAPI could bind to the surface of micron hydroxyapatite (μ HAp) through physical adsorption, thus they employed DAPI staining to reconstruct μ HAp distribution in hybrid cross-linked membranous scaffold (HCLS). If the view can stand, how can DAPI staining be used to distinguish cells from HAp in mature bone. Besides, the added reference 29 (Shao et al., *Adv Mater*, 2018, 30: 1704876.) did not cover any details of physical adsorption between DAPI and μ HAp.

2. The updated references 30 (Holten-Andersen et al., *Langmuir*, 2009, 25: 3323-3326.) and 31 (Ryu et al., *Adv Funct Mater*, 2010, 20: 2132-2139.) were about adhesion of polydopamine, they inadequately supported the conclusion that the incorporation of μ HAp could increase cross-linking degree and strengthen internal network structure.

3. The authors attributed the decrease of BV/TV value and new bone volume of the Blank group at 12 weeks in beagle dog model to autophagy imbalance and bone-resorption of active osteoclasts. Unfortunately, the viewpoint could not convince the reviewer. What exactly does "lower stress level of Blank group" mean? In addition, no experimental result showed that osteoclasts were more active than osteoblasts in the Blank group.

4. The error bar values in Fig.1E3 were obviously much smaller than those in its original drawing (Fig.1F in the original manuscript). Please confirm whether the data were represented as means \pm SD.

Dear Editor and Referees,

We highly appreciate the valuable comments and suggestions of reviewers, which greatly helped us to improve the quality of our manuscript. According to these comments and suggestions, we revised the manuscript carefully.

We hereby, on behalf of the co-authors, submitted the revised manuscript entitled “Instantly fixable and self-adaptive scaffold for skull regeneration by autologous stem cell recruitment and angiogenesis” to “**Nature Communications**” for your consideration, which included the Manuscript (8 Figures and 1 Scheme) and Supporting Information (31 Figures, 2 Videos and 2 Tables). The source data underlying Figure 1-8 and Supplementary Figure 13, 14 and 30C were provided as a Source Data file.

The point-to-point response to the reviewers’ comments was listed as follows:

Response to Reviewer #1 (Remarks to the Author):

A Review of NCOMMS-21-12583A

Title: Instantly Fixable and Self-adaptive Membranous Scaffold for Skull Regeneration by Autologous Stem Cell Recruitment and Angiogenesis

Q1: *Even after the extensive evaluation, the revised manuscript cannot be recommended for the publication. All reviewers questioned the novelty of the present study, the principle of material design, and related scientific rationales, specifically as compared with a number of previous studies using (1) dopamine modification, (2) conventional polymeric & inorganic composites for (3) skull bone regeneration. In spite of the additional transcriptomic & immunological analysis, the major drawback of the current study is still insufficient verification of novelty in materials design along with discrete significances. Only Figure 1A describes a brief chemistry for incorporation of hyaluronic acid and hydroxyapatite, with the aid of dopamine and calcium chelating. However, it is still ambiguous to confirm this approach and resulted material design is superior for the publication in Nature Communications.*

Reply: Thanks for your valuable comment that let us know what the deficiencies of the previous manuscript were. According to this suggestive comment, we conducted additional experiments to

clarify the superiority of this material.

Bone is a highly mineralized and organic/inorganic hybridized composites in which aligned organic type I collagen (Col I) fibers was embedded with inorganic hydroxyapatite (HAp) as its primary framework elements. Currently, recreating natural bony matrix structures by integrating collagen fibrils and HAp is a very attractive prospect. In this manuscript, dopamine-modified hyaluronic acid (HAD) was introduced as a “bridge” to chelate Ca^{2+} derived from μHAp and bound type I collagen (Col I) by Michael addition reaction (Fig. 1). These individual ingredients might not be provided new advances in bone regeneration, but the integration of HAD, Col I and μHAp presented structural superiority and satisfactory biological functions, and was rarely reported.

Specifically, this strategy effectively integrated organic and inorganic phases with strong chemical connection in molecular level, which mimicked the construction of natural bony ECM, and endowed with satisfactory biological functions to initiate the process of bone regeneration without exogenous cells and factors. Meanwhile, these individual bony matrix components had been widely used in clinical applications, suggesting acceptable long-term biocompatibility, which was vital as implantable biomaterial. As a result, the obtained hybrid cross-linked scaffold (HCLS) could maintain structural integrity through flexible deformation to respond external stress in the defect, and present well mechanical match and interface integration with bony host environment by tissue adhesion. Furthermore, the proper immune microenvironment provided by HCLS could regulated macrophages polarization to M2 phenotype. After implantation in both rabbit and beagle dog cranial defect models, HCLS could rapidly initiate angiogenesis and osteogenesis by *in-situ* recruitment of ESCs into defect sites, and subsequently accelerate osteo-differentiation to regenerate skull.

For further demonstrating the structural superiority of HCLS scaffold, additional experiments by XPS and SEM analysis were performed to reveal chemical bonding and calcium chelation at the molecular level. The chemical bonding between HAD and Col I was further investigated by XPS (Fig. 1A). Higher -C=O (284.3 eV, 62.2% and 530.8 eV, 45.7%) and lower -C-OH peaks area (286.2 eV, 30.4% and 532.6 eV, 19.2%) were observed in DCLS as compared to HAD (-C=O, 284.3eV, 38.5% and 530.8eV, 23.1% ; -C-OH, 286.2 eV, 60.8% and 532.6 eV, 72.3%), indicating

that HAD was oxidized and bonded with amino groups in Col I (**Fig. 1B-C**). Furthermore, relatively more -C=O in DCLS (284.3 eV, 62.2%) suggested that phenolic hydroxyl groups in HAD were converted to quinone groups during oxidation, which might provide more nucleation sites and lead to nucleation and growth of hydroxyapatite crystals. Calcium chelation analysis in HCLS demonstrated obvious Ca2p peak (Ca2p3/2, 346.5 eV), which was different from HAp (Ca2p3/2, 346.8 eV). The disappeared P2p peak in HCLS suggested that Ca²⁺ originated from HAp, which was ionized to form chelated calcium (coordinated calcium) with cross-linked DCLS polymer (**Fig. 2A-C**).

In order to confirm chelation capacity of calcium ion, DCLS and HCLS were treated with 0.3 M CaCl₂ and characterized by X-ray photoelectron spectroscopy (**Fig. 3A**). There were obvious Ca2p peak (Ca2p1/2, 351.0 eV and Ca2p3/2, 347.5 eV) in HCLS-0.3M Ca²⁺, indicating sustaining calcium ion chelation ability of HCLS, even after chelating ionized calcium in the surface of HAp (**Fig. 3B**). Besides, more -C=O (284.3 eV and 531.3 eV) was found in comparison with -C-OH (286.2 eV and 532.9 eV) both in DCLS and HCLS, implying that catechol groups were oxidized to quinones which could further chelated with Ca²⁺ (**Fig. 3C and D**). The *in vitro* mineralization experiments were carried out to further verify chelation capacity of calcium ion in HCLS (**Fig. 4**). SEM images suggested that micro-spherical osteoid apatite agglomerates were dispersed on the surface of HCLS, high-magnification SEM images revealed that all the osteoid apatite particles had nano-lath-like structure, a typical hydroxyapatite crystal (**Fig. 4A**).²⁹ EDS analysis demonstrated that crystals were mainly composed of Ca and P element with Ca/P ratio at 1.74, which was closest to mineralized hydroxyapatite in natural bone (1.67) (**Fig. 4B and C**).³⁰

Next, SEM revealed the *in vivo* mineralization at the interface between scaffold and host bone after 7 and 14 day implantation (**Fig. 5**). More obviously mineralized fiber structure in HCLS was observed on day 7, and the mineralized matrix became more intense while nano-lath-like structural osteoid apatite particles appeared in HCLS on day 14, which was consistent with the results of *in vitro* mineralization (**Fig. 5A**). Ca/P ratios and element analysis results further confirmed the minerals in HCLS was closer to natural HAp (**Fig. 5B**). These results suggested that HCLS had strong calcium binding ability, which could induce HAp formation to facilitate good osseointegration. The corresponding descriptions have been added to the revised manuscript and

were marked as red.

Fig. 1 Chemical bonding between HAD and Col I based on X-ray photoelectron spectroscopy. A) Full x-ray photoelectron spectroscopy spectra of HAD and DCLS. B) C) High-resolution x-ray photoelectron spectroscopy spectra of C1s and O1s in HAD and DCLS.

Fig. 2 Calcium chelation analysis in HCLS based on X-ray photoelectron spectroscopy. A) Full x-ray photoelectron spectroscopy spectra of HAp and HCLS. B) C) High-resolution x-ray photoelectron spectroscopy spectra of Ca2p and P2p in HAp and HCLS.

Fig. 3 Calcium ion chelating capacity analysis based on x-ray photoelectron spectroscopy spectra. A) Full x-ray photoelectron spectroscopy spectra of DCLS and HCLS treated with 0.3M CaCl₂ solution. B) C) D) High-resolution x-ray photoelectron spectroscopy spectra of Ca2p, C1s and O1s in DCLS-0.3M Ca²⁺ and HCLS-0.3M Ca²⁺.

Fig. 4 Calcium chelation and mineralization of HCLS in SBF at 37°C for 7 days. A) SEM image of bone-like apatite at different magnifications in HCLS. B) EDS evaluation of Ca/P ratios. C) EDS mapping of calcium and phosphate element in HCLS after 7 days mineralization.

Fig. 5 Evaluation of mineralization at the interface between implant and host bone. A) SEM images of the implants. B) Ca/P ratios and element analysis based on EDS.

Response to Reviewer #2 (Remarks to the Author):

Q1. Generally the authors have addressed all my scientific comments and the manuscript is improved by the inclusion of additional data. However, the language in paper still needs careful editing as is not sufficiently fluent in English - and the new text added is particularly poor. If this is addressed, I'm happy to recommend publication.

Reply: Thanks for your valuable comments. According to your suggestion, we had carefully edited the language in whole paper, and the revised part had been marked as red.

Response to Reviewer #3 (Remarks to the Author):

The authors added several experiments to support their conclusions, and the manuscript was improved, but there are details that remain poorly addressed.

*Q1. The authors stated that DAPI could bind to the surface of micron hydroxyapatite (μ HAp) through physical adsorption, thus they employed DAPI staining to reconstruct μ HAp distribution in hybrid cross-linked membranous scaffold (HCLS). If the view can stand, how can DAPI staining be used to distinguish cells from HAp in mature bone. Besides, the added reference 29 (Shao et al., *Adv Mater*, 2018, 30: 1704876.) did not cover any details of physical adsorption between DAPI and μ HAp.*

Reply: Thanks for your valuable comments. The distribution of μ HAp in HCLS was reconstructed by calcein staining in revised manuscript, which was a good fluorescent indicator of calcium complex.^{6,31,32} The CLSM confirmed μ HAp was uniformly distributed in HCLS. The corresponding descriptions have been added to the revised manuscript and were marked as red.

6. Luo, Z. et al. Injectable 3D porous micro-scaffolds with a bio-engine for cell transplantation and tissue regeneration. *Adv. Funct. Mater.* **28**, 1804335 (2018).

31. Shao, C. Y., et al. Citrate improves collagen Mineralization via interface wetting: A physicochemical understanding of biomineralization control. *Adv. Mater.* **30**, 1704876 (2018).

32. Hale, L. V., Ma, Y. F., Santerre, R. F. Semi-quantitative fluorescence analysis of calcein binding as a measurement of in vitro mineralization. *Calcif. Tissue Int.* **67**, 80-84 (2000).

Fig. 6 CLSM image of μ HAp distribution by calcein staining.

Q2. The updated references 30 (Holten-Andersen et al., Langmuir, 2009, 25: 3323-3326.) and 31 (Ryu et al., Adv Funct Mater, 2010, 20: 2132-2139.) were about adhesion of polydopamine, they inadequately supported the conclusion that the incorporation of μ HAp could increase cross-linking degree and strengthen internal network structure.

Reply: Thanks for your valuable comments. For further demonstrating the structural superiority of HCLS scaffold, additional experiments by XRD and SEM analysis were performed to reveal chemical bonding and calcium chelation at the molecular level. The chemical bonding between HAD and Col I was further investigated by X-ray photoelectron spectroscopy (**Fig. 1A**). Higher -C=O (284.3 eV, 62.2% and 530.8 eV, 45.7%) and lower -C-OH peaks area (286.2 eV, 30.4% and 532.6 eV, 19.2%) were observed in DCLS as compared to HAD (-C=O, 284.3eV, 38.5% and 530.8eV, 23.1%) and (-C-OH, 286.2 eV, 60.8% and 532.6 eV, 72.3%), indicating that HAD was oxidized and bonded with amino groups in Col I (**Fig. 1B-C**). Furthermore, relatively more -C=O in DCLS (284.3 eV, 62.2%) suggested that phenolic hydroxyl groups in HAD were converted to quinone groups during oxidation, which might provide more nucleation sites and lead to nucleation and growth of hydroxyapatite crystals. Calcium chelation analysis in HCLS demonstrated obvious Ca2p peak (Ca2p3/2, 346.5 eV), which was different from HAp (Ca2p3/2, 346.8 eV). The disappeared P2p peak in HCLS suggested that Ca²⁺ originated from HAp, which was ionized to form chelated calcium (coordinated calcium) with cross-linked DCLS polymer (**Fig. 2A-C**). These results suggested that the incorporation of μ HAp might increase cross-linking degree by newly formed chelation between Ca²⁺ derived from μ HAp and DCLS polymer to strengthen internal network structure. The corresponding descriptions have been added to the revised manuscript and were marked as red.

Fig. 1 Chemical bonding between HAD and Col I based on X-ray photoelectron spectroscopy. A) Full x-ray photoelectron spectroscopy spectra of HAD and DCLS. B) C) High-resolution x-ray photoelectron spectroscopy spectra of C1s and O1s in HAD and DCLS.

Fig. 2 Calcium chelation analysis in HCLS based on X-ray photoelectron spectroscopy. A) Full x-ray photoelectron spectroscopy spectra of HAp and HCLS. B) C) High-resolution x-ray photoelectron spectroscopy spectra of Ca2p and P2p in HAp and HCLS.

Q3. The authors attributed the decrease of BV/TV value and new bone volume of the Blank group at 12 weeks in beagle dog model to autophagy imbalance and bone-resorption of active osteoclasts. Unfortunately, the viewpoint could not convince the reviewer. What exactly does “lower stress level of Blank group” mean? In addition, no experimental result showed that osteoclasts were more active than osteoblasts in the Blank group.

Reply: Thanks for your valuable comments. The unreasonable explanation about the decrease of BV/TV value and new bone volume of Blank group has been corrected in revised manuscript, the possible reasons might be as follows:

First, because of the large size of defect area in beagle dog model, it was difficult to accurately drill a 15 mm diameter circle by using the high-speed ball milling drill in the operation process. Therefore, the actual defect diameter might be greater than 15 mm, and the defect sizes among samples had unavoidable deviation (**Supplementary Fig. 26A**).

Secondly, it has been reported that high speed sawing or drilling has obvious thermal damage to local cortical bone clinically.⁵⁸⁻⁶¹ The high-speed ball milling drill inevitably brought about thermal damage to surrounding cortical bone, which might result in local osteonecrosis, and subsequent bone resorption in necrotic bone. The combination of these factors might cause the enlargement of defect area at 12 weeks in beagle dog model.

58. Singh, T. P., Yusoff, A. H., & Chian, Y. K. How safe is high-speed burring in spine surgery? An *in vitro* study on the effect of rotational speed and heat generation in the bovine spine. *Spine*. **40**, 866-72 (2015).

59. Li, S., Shu, C., & Brånemark, P. I. Heat shock-induced necrosis and apoptosis in osteoblasts. *J. Orthop. Res.* **17**, 891-899 (1999).

60. Steven, L., Haddad, Andrew, R., & Hsu, et al. Effects of continuous irrigation during burring on thermal necrosis and fusion strength in a rabbit arthrodesis model. *Foot Ankle Int.* **35**, 796-801 (2014).

61. Dolan, E. B., Haugh, M. G., D Tallon, Casey, C., & Mcnamara, L. M. Heat-shock-induced cellular responses to temperature elevations occurring during orthopedic cutting. *J. R. Soc. Interface.* **9**, 3503-3513 (2012).

Q4. The error bar values in Fig.1E3 were obviously much smaller than those in its original drawing (Fig.1F in the original manuscript). Please confirm whether the data were represented as means ± SD.

Reply: Thanks for your valuable comments. We apologized for making such mistakes. According to your suggestion, we had carefully checked and confirmed error bar values in Fig.1E3 according to original data, which was represented as means \pm SD (Fig.1F in the original manuscript). The modified Fig.1E3 was remarked as new Fig. 1F3 in revised manuscript.

REVIEWERS' COMMENTS

Reviewer #3 (Remarks to the Author):

The authors have addressed most of the reviewers' questions and the manuscript is improved. However, their explanation of the decrease of BV/TV value and new bone volume of the Blank group at 12 weeks in beagle dog model still cannot convince the reviewer. The defect diameters among samples could be in a controllable error range even using a high-speed ball milling drill in the operation process. Besides, the heat generated by high-speed ball milling drill could be attenuated by irrigation with saline solution. More operational details during surgical should be added to the method.

Dear Editor and Referees,

We highly appreciate the valuable comments and suggestions of reviewers, which greatly helped us to improve the quality of our manuscript. According to these comments and suggestions, we revised the manuscript carefully.

We hereby, on behalf of the co-authors, submitted the revised manuscript entitled “Instantly fixable and self-adaptive scaffold for skull regeneration by autologous stem cell recruitment and angiogenesis” to “**Nature Communications**” for your consideration, which included the Manuscript (8 Figures and 1 Scheme) and Supporting Information (31 Figures, 2 Videos and 2 Tables). The source data underlying Figure 1-8 and Supplementary Figure 13, 14 and 30C were provided as a Source Data file.

The point-to-point response to the reviewers’ comments was listed as follows:

Response to Reviewer #3 (Remarks to the Author):

A Review of NCOMMS-21-12583B

Title: Instantly Fixable and Self-adaptive Membranous Scaffold for Skull Regeneration by Autologous Stem Cell Recruitment and Angiogenesis

Q1 : *The authors have addressed most of the reviewers' questions and the manuscript is improved. However, their explanation of the decrease of BV/TV value and new bone volume of the Blank group at 12 weeks in beagle dog model still cannot convince the reviewer. The defect diameters among samples could be in a controllable error range even using a high-speed ball milling drill in the operation process. Besides, the heat generated by high-speed ball milling drill could be attenuated by irrigation with saline solution. More operational details during surgical should be added to the method.*

Reply: Thanks for your valuable comments. According to your suggestion, the operational details during surgical procedure were described as follows:

After anesthetizing the beagle dogs, the skull was marked with a marker. The flaps or muscle flaps were carefully separated by high-frequency electric knife to avoid cerebrospinal fluid

leakage or blade injury to brain tissue, so as to reduce the occurrence of subcutaneous dural effusion after operation. Initial hemostasis was performed with medical gauze and the wound was cleaned with normal saline. A spreader was then used to support both sides of the incision to expose the location of hole. Then a hand-held dental ball drill ($\Phi = 2.3$ mm, HM1023-round head, Rick ward, Corporation, China) was used to drill holes with a diameter of about 15 mm on both sides of the skull. During the drilling process, saline solution was rinsed every 2 minutes or so to remove dross and exudate blood while cooling down. Initial hemostasis was performed with medical gauze, cerebral cotton tablets were covered on the defect, and physiological saline was added to make it wet.

Although the heat generated by high-speed ball milling drill could be attenuated by irrigation with saline solution, a 2-minutes interval of saline flushing might not be sufficient for minimizing thermal damage caused by high-speed ball milling drill. We would consider reducing the interval to 1 minutes in future experiments. The corresponding descriptions have been added to revised manuscript and were marked as red.

I expect your comments on this revised manuscript at your earliest convenience.

Thanks and all the best.

Yours sincerely,

Yong Sun, Ph. D and Yujiang Fan, Ph. D

Signature: 
National Engineering Research Center for Biomaterials

Sichuan University

Chengdu, China